# Regret Bounds for Robust Adaptive Control of the Linear Quadratic Regulator

**Sarah Dean**    **Horia Mania**    **Nikolai Matni**    **Benjamin Recht**    **Stephen Tu**

University of California, Berkeley

## Abstract

We consider adaptive control of the Linear Quadratic Regulator (LQR), where an unknown linear system is controlled subject to quadratic costs. Leveraging recent developments in the estimation of linear systems and in robust controller synthesis, we present the first provably polynomial time algorithm that provides high probability guarantees of sub-linear regret on this problem. We further study the interplay between regret minimization and parameter estimation by proving a lower bound on the expected regret in terms of the exploration schedule used by any algorithm. Finally, we conduct a numerical study comparing our robust adaptive algorithm to other methods from the adaptive LQR literature, and demonstrate the flexibility of our proposed method by extending it to a demand forecasting problem subject to state constraints.

## 1    Introduction

The problem of adaptively controlling an unknown dynamical system has a rich history, with classical asymptotic results of convergence and stability dating back decades [12, 13]. Of late, there has been a renewed interest in the study of a particular instance of such problems, namely the adaptive Linear Quadratic Regulator (LQR), with an emphasis on *non-asymptotic* guarantees of stability and performance. Initiated by Abbasi-Yadkori and Szepesvári [1], there have since been several works analyzing the regret suffered by various adaptive algorithms on LQR– here the regret incurred by an algorithm is thought of as a measure of deviations in performance from optimality over time. These results can be broadly divided into two categories: those providing high-probability guarantees for a single execution of the algorithm [1, 4, 8, 11], and those providing bounds on the expected *Bayesian* regret incurred over a family of possible systems [2, 16]. As we discuss in more detail, these methods all suffer from one or several of the following limitations: restrictive and unverifiable assumptions, limited applicability, and computationally intractable subroutines. In this paper, we provide, to the best of our knowledge, the first polynomial-time algorithm for the adaptive LQR problem that provides high probability guarantees of sub-linear regret, and that does not require unverifiable or unrealistic assumptions.

**Related Work.**    There is a rich body of work on the estimation of linear systems as well as on the robust and adaptive control of unknown systems. We target our discussion to works on non-asymptotic guarantees for the LQR control of an unknown system, broadly divided into three categories.

*Offline estimation and control synthesis:* In a non-adaptive setting, i.e., when system identification can be done offline prior to controller synthesis and implementation, the first work to provide end-to-end guarantees for the LQR optimal control problem is that of Fiechter [10], who shows that the *discounted* LQR problem is PAC-learnable. Dean et al. [6] improve on this result, and provide the first end-to-end sample complexity guarantees for the infinite horizon average cost LQR problem.

*Optimism in the Face of Uncertainty (OFU):* Abbasi-Yadkori and Szepesvári [1], Faradonbeh et al. [8], and Ibrahimi et al. [11] employ the *Optimism in the Face of Uncertainty* (OFU) principle [5], which optimistically selects model parameters from a confidence set by choosing those that lead to the *best* closed-loop (infinite horizon) control performance, and then plays the corresponding optimal controller, repeating this process online as the confidence set shrinks. While OFU in the LQR setting has been shown to achieve optimal regret $\widetilde{\mathcal{O}}(\sqrt{T})$, its implementation requires solving a non-convex optimization problem to precision $\widetilde{\mathcal{O}}(T^{-1/2})$, for which no provably efficient implementation exists.

*Thompson Sampling (TS):* To circumvent the computational roadblock of OFU, recent works replace the intractable OFU subroutine with a random draw from the model uncertainty set, resulting in *Thompson Sampling* (TS) based policies [2, 4, 16]. Abeille and Lazaric [4] show that such a method achieves $\widetilde{\mathcal{O}}(T^{2/3})$ regret with high-probability for scalar systems. However, their proof does not extend to the non-scalar setting. Abbasi-Yadkori and Szepesvári [2] and Ouyang et al. [16] consider expected regret in a Bayesian setting, and provide TS methods which achieve $\widetilde{\mathcal{O}}(\sqrt{T})$ regret. Although not directly comparable to our result, we remark on the computational challenges of these algorithms. Whereas the proof of Abbasi-Yadkori and Szepesvári [2] was shown to be incorrect [15], Ouyang et al. [16] make the restrictive assumption that there exists a (known) initial compact set $\Theta$ describing the uncertainty in the system parameters, such that for any system $\theta_1 \in \Theta$, the optimal controller $K(\theta_1)$ is stabilizing when applied to any other system $\theta_2 \in \Theta$. No means of constructing such a set are provided, and there is no known tractable algorithm to verify if a given set satisfies this property. Also, it is implicitly assumed that projecting onto this set can be done efficiently.

**Contributions.** To develop the first polynomial-time algorithm that provides high probability guarantees of sub-linear regret, we leverage recent results from the estimation of linear systems [17], robust controller synthesis [14, 19], and coarse-ID control [6]. We show that our robust adaptive control algorithm: (i) guarantees stability and near-optimal performance at all times; (ii) achieves a regret up to time $T$ bounded by $\widetilde{\mathcal{O}}(T^{2/3})$; and (iii) is based on finite-dimensional semidefinite programs of size logarithmic in $T$.

Furthermore, our method estimates the system parameters at $\widetilde{\mathcal{O}}(T^{-1/3})$ rate in operator norm. Although system parameter identification is not necessary for optimal control performance, an accurate system model is often desirable in practice. Motivated by this, we study the interplay between regret minimization and parameter estimation, and identify fundamental limits connecting the two. We show that the expected regret of our algorithm is lower bounded by $\Omega(T^{2/3})$, proving that our analysis is sharp up to logarithmic factors. Moreover, our lower bound suggests that the estimation rate achievable by any algorithm with $\mathcal{O}(T^\alpha)$ regret is $\Omega(T^{-\alpha/2})$.

Finally, we conduct a numerical study of the adaptive LQR problem, in which we implement our algorithm, and compare its performance to heuristic implementations of OFU and TS based methods. We show on several examples that the regret incurred by our algorithm is comparable to that of the OFU and TS based methods. Furthermore, the infinite horizon cost achieved by our algorithm at any given time on the true system is consistently lower than that attained by OFU and TS based algorithms. Finally, we use a demand forecasting example to show how our algorithm naturally generalizes to incorporate environmental uncertainty and safety constraints. The full version of this paper is [7].

## 2   Problem Statement and Preliminaries

In this work we consider adaptive control of the following discrete-time linear system

$$x_{k+1} = A_\star x_k + B_\star u_k + w_k \; , \;\; w_k \overset{\text{i.i.d.}}{\sim} \mathcal{N}(0, \sigma_w^2 I) \; , \tag{2.1}$$

where $x_k \in \mathbb{R}^n$ is the state, $u_k \in \mathbb{R}^p$ is the control input, and $w_k \in \mathbb{R}^n$ is the process noise. We assume that the state variables are observed exactly and, for simplicity, that $x_0 = 0$. We consider the *Linear Quadratic Regulator* optimal control problem, given by cost matrices $Q \succeq 0$ and $R \succ 0$,

$$J_\star = \min_u \lim_{T \to \infty} \frac{1}{T} \mathbb{E} \left[ \sum_{k=1}^{T} x_k^\top Q x_k + u_k^\top R u_k \right] \text{ s.t. dynamics (2.1)} \;, \tag{2.2}$$

where the minimum is taken over measurable functions $u = \{u_k(\cdot)\}_{k \geq 1}$, with each $u_k$ adapted to the history $x_k, x_{k-1}, \ldots, x_1$, and possibe additional randomness independent of future states. Given knowledge of $(A_\star, B_\star)$, the optimal policy is a static state-feedback law $u_k = K_\star x_k$, where $K_\star$ is derived from the solution to a discrete algebraic Riccati equation.

We are interested in algorithms which operate without knowledge of the true system transition matrices $(A_\star, B_\star)$. We measure the performance of such algorithms via their regret, defined as

$$\mathsf{Regret}(T) := \sum_{k=1}^{T} (x_k^\top Q x_k + u_k^\top R u_k - J_\star) . \tag{2.3}$$

The regret of any algorithm is lower-bounded by $\Omega(\sqrt{T})$, a bound matched by OFU up to logarithmic factors [8]. However, after each epoch, OFU requires optimizing a non-convex objective to $\mathcal{O}(T^{-1/2})$ precision. Instead, our method uses a subroutine based on quasi-convex optimization and robust control.

## 2.1 Preliminaries: System Level Synthesis

We briefly describe the necessary background on robust control and System Level Synthesis [19] (SLS). These tools were recently used by Dean et al. [6] to provide non-asymptotic bounds for LQR in the offline "estimate-and-then-control" setting. In the appendix of the full version [7] we expand on these preliminaries.

Consider the dynamics (2.1), and fix a static state-feedback control policy $K$, i.e., let $u_k = K x_k$. Then, the closed loop map from the disturbance process $\{w_0, w_1, \ldots\}$ to the state $x_k$ and control input $u_k$ at time $k$ is given by

$$\begin{array}{rcl} x_k & = & \sum_{t=1}^{k} (A_\star + B_\star K)^{k-t} w_{t-1} , \\ u_k & = & \sum_{t=1}^{k} K (A_\star + B_\star K)^{k-t} w_{t-1} . \end{array} \tag{2.4}$$

Letting $\Phi_x(k) := (A_\star + B_\star K)^{k-1}$ and $\Phi_u(k) := K(A_\star + B_\star K)^{k-1}$, we can rewrite Eq. (2.4) as

$$\begin{bmatrix} x_k \\ u_k \end{bmatrix} = \sum_{t=1}^{k} \begin{bmatrix} \Phi_x(k-t+1) \\ \Phi_u(k-t+1) \end{bmatrix} w_{t-1} , \tag{2.5}$$

where $\{\Phi_x(k), \Phi_u(k)\}$ are called the *closed loop system response elements* induced by the controller $K$. The SLS framework shows that for any elements $\{\Phi_x(k), \Phi_u(k)\}$ constrained to obey

$$\Phi_x(k+1) = A_\star \Phi_x(k) + B_\star \Phi_u(k) , \;\; \Phi_x(1) = I , \;\; \forall k \geq 1 , \tag{2.6}$$

there exists some controller that achieves the desired system responses (2.5). The state-feedback parameterization result in Theorem 1 of Wang et al. [19] formalizes this observation: the SLS framework therefore allows for any optimal control problem over linear systems to be cast as an optimization problem over elements $\{\Phi_x(k), \Phi_u(k)\}$, constrained to satisfy the affine equations (2.6). Comparing equations (2.4) and (2.5), we see that the former is non-convex in the controller $K$, whereas the latter is affine in the elements $\{\Phi_x(k), \Phi_u(k)\}$, enabling solutions to previously difficult optimal control problems.

As we work with infinite horizon problems, it is notationally more convenient to work with *transfer function* representations of the above objects, which can be obtained by taking a $z$-transform of their time-domain representations. The frequency domain variable $z$ can be informally thought of as the time-shift operator, i.e., $z\{x_k, x_{k+1}, \ldots\} = \{x_{k+1}, x_{k+2}, \ldots\}$, allowing for a compact representation of LTI dynamics. We use boldface letters to denote such transfer functions, e.g., $\mathbf{\Phi}_x(z) = \sum_{k=1}^{\infty} \Phi_x(k) z^{-k}$. Then, the constraints (2.6) can be rewritten as

$$[zI - A_\star \quad -B_\star] \begin{bmatrix} \mathbf{\Phi}_x \\ \mathbf{\Phi}_u \end{bmatrix} = I , \tag{2.7}$$

and the corresponding (not necessarily static) control law $\mathbf{u} = \mathbf{K}\mathbf{x}$ is given by $\mathbf{K} = \mathbf{\Phi}_u \mathbf{\Phi}_x^{-1}$.

Although other approaches to optimal controller design exists, we argue now that the SLS parameterization has some appealing properties when applied to the control of uncertain systems. In particular,

suppose that rather than having access to the true system transition matrices $(A_\star, B_\star)$, we instead only have access to estimates $(\widehat{A}, \widehat{B})$. The SLS framework allows us to characterize the system responses achieved by a controller, computed using only the estimates $(\widehat{A}, \widehat{B})$, on the true system $(A_\star, B_\star)$. Specifically, if we denote $\widehat{\boldsymbol{\Delta}} := (\widehat{A} - A_\star)\boldsymbol{\Phi}_x + (\widehat{B} - B_\star)\boldsymbol{\Phi}_u$, simple algebra shows that

$$\begin{bmatrix} zI - \widehat{A} & -\widehat{B} \end{bmatrix} \begin{bmatrix} \boldsymbol{\Phi}_x \\ \boldsymbol{\Phi}_u \end{bmatrix} = I \quad \text{if and only if} \quad \begin{bmatrix} zI - A_\star & -B_\star \end{bmatrix} \begin{bmatrix} \boldsymbol{\Phi}_x \\ \boldsymbol{\Phi}_u \end{bmatrix} = I + \widehat{\boldsymbol{\Delta}} .$$

The robust stability result in Theorem 2 of Matni et al. [14] shows that if $(I + \widehat{\boldsymbol{\Delta}})^{-1}$ exists, then the controller $\mathbf{K} = \boldsymbol{\Phi}_u \boldsymbol{\Phi}_x^{-1}$, computed using only the estimates $(\widehat{A}, \widehat{B})$, achieves the following response on the true system $(A_\star, B_\star)$:

$$\begin{bmatrix} \mathbf{x} \\ \mathbf{u} \end{bmatrix} = \begin{bmatrix} \boldsymbol{\Phi}_x \\ \boldsymbol{\Phi}_u \end{bmatrix} (I + \widehat{\boldsymbol{\Delta}})^{-1} \mathbf{w} .$$

Further, if $\mathbf{K}$ stabilizes the system $(\widehat{A}, \widehat{B})$, and $(I + \widehat{\boldsymbol{\Delta}})^{-1}$ is stable (simple sufficient conditions can be derived to ensure this, see [6]), then $\mathbf{K}$ is also stabilizing for the true system. It is this transparency between system uncertainty and controller performance that we exploit in our algorithm.

We end this discussion with the definition of a function space that we use extensively throughout:

$$\mathcal{S}(C, \rho) := \left\{ \mathbf{M} = \sum_{k=1}^{\infty} M(k) z^{-k} \mid \|M(k)\| \leq C\rho^k , \ k = 1, 2, ... \right\} . \qquad (2.8)$$

The space $\mathcal{S}(C, \rho)$ consists of (strictly proper) stable transfer functions that satisfy a certain decay rate in the spectral norm of their impulse response elements. We denote the restriction of $\mathcal{S}(C, \rho)$ to the space of $F$-length finite impulse response (FIR) filters by $\mathcal{S}_F(C, \rho)$, i.e., $\mathbf{M} \in \mathcal{S}_F(C, \rho)$ if $\mathbf{M} \in \mathcal{S}(C, \rho)$, and $M(k) = 0$ for all $k > F$.

We equip $\mathcal{S}(C, \rho)$ with the $\mathcal{H}_\infty$ and $\mathcal{H}_2$ norms, which are infinite horizon analogs of the spectral and Frobenius norms of a matrix, respectively: $\|\mathbf{M}\|_{\mathcal{H}_\infty} = \sup_{\|\mathbf{w}\|_2 = 1} \|\mathbf{M}\mathbf{w}\|_2$ and $\|\mathbf{M}\|_{\mathcal{H}_2} = \sqrt{\sum_{k=1}^{\infty} \|M(k)\|_F^2}$. The $\mathcal{H}_\infty$ and $\mathcal{H}_2$ norm have distinct interpretations. The $\mathcal{H}_\infty$ norm of a system $\mathbf{M}$ is equal to its $\ell_2 \mapsto \ell_2$ operator norm, and can be used to measure the robustness of a system to unmodelled dynamics [20]. The $\mathcal{H}_2$ norm has a direct interpretation as the energy transferred to the system by a white noise process, and is hence closely related to the LQR optimal control problem. Unsurprisingly, the $\mathcal{H}_2$ norm appears in the objective function of our optimization problem, whereas the $\mathcal{H}_\infty$ norm appears in the constraints to ensure robust stability and performance.

## 3 Algorithm and Guarantees

Our proposed robust adaptive control algorithm for LQR is shown in Algorithm 1. We note that while Line 8 of Algorithm 1 is written as an infinite-dimensional optimization problem, it can be formulated in terms of finite-dimensional decision variables $\{\Phi_x(k), \Phi_u(k)\}_{k=1}^F$ due to the restriction to FIR filters. In this formulation, the $\mathcal{H}_2$ cost can be written as a Frobenius norm and the $\mathcal{H}_\infty$ constraint reduces to a linear matrix inequality. Therefore, the inner optimization can be equivalently written as a semidefinite program over $\mathcal{O}(F_i(n^2 + np))$ decision variables. We describe this transformation in detail in appendix Section G of the full version [7]. We also note that the outer optimization over $\gamma$ can be performed efficiently by bisection search because the objective is jointly quasi-convex in the decision variables and is smooth with respect to $\gamma$ in the feasible domain.

Some remarks on practice are in order. First, in Line 6, only the trajectory data collected during the $i$-th epoch is used for the least squares estimate. Second, the epoch lengths we use grow exponentially in the epoch index. These settings are chosen primarily to simplify the analysis; in practice all the data collected should be used, and it may be preferable to use a slower growing epoch schedule (such as $T_i = C_T(i + 1)$). Additionally, for storage considerations, instead of performing a batch least squares update of the model, a recursive least squares (RLS) estimator rule can be used to update the parameters in an online manner. Furthermore, many constants in Algorithm 1 depend on the unknown system to be consistent with our data-independent analysis. In practice, these parameters can be estimated from collected data.

Finally, we note that the proofs for all results in this section can be found in the full version [7].

---

**Algorithm 1** Robust Adaptive Control Algorithm

---

**Require:** Stabilizing controller $\mathbf{K}^{(0)}$, failure probability $\delta \in (0, 1)$, and constants $(C_\star, \rho_\star, \|K_\star\|)$.

1: Set $C_x \leftarrow \frac{\mathcal{O}(1)C_\star}{(1-\rho_\star)^3}$, $C_u \leftarrow \|K_\star\|C_x$, and $\rho \leftarrow .999 + .001\rho_\star$.

2: Set $C_T \leftarrow \widetilde{\mathcal{O}}\left((n+p)\frac{C_\star^4(1+\|K_\star\|)^4}{(1-\rho_\star)^8}\right)$.

3: **for** $i = 0, 1, 2, ...$ **do**

4:     Set $T_i \leftarrow C_T 2^i$ and $\sigma_{\eta,i}^2 \leftarrow \sigma_w^2 (T_i/C_T)^{-1/3}$.

5:     Set $D_i = \{(x_k^{(i)}, u_k^{(i)})\}_{k=1}^{T_i} \leftarrow$ evolve system forward $T_i$ steps, where each action $u_k^{(i)}$ is obtained from the controller $\mathbf{K}^{(i)}$ plus an additional noise term for exploration. More precisely, $\mathbf{u}^{(i)} = \mathbf{K}^{(i)}\mathbf{x}^{(i)} + \boldsymbol{\eta}^{(i)}$, where each entry of $\boldsymbol{\eta}^{(i)}$ is drawn i.i.d. from $\mathcal{N}(0, \sigma_{\eta,i}^2 I_p)$.

6:     Set $(\widehat{A}_i, \widehat{B}_i) \leftarrow \arg\min_{A,B} \sum_{k=1}^{T_i-1} \frac{1}{2}\|x_{k+1}^{(i)} - Ax_k^{(i)} - Bu_k^{(i)}\|_2^2$.

7:     Set $\varepsilon_i \leftarrow \widetilde{\mathcal{O}}\left(\frac{\sigma_w\|K_\star\|C_\star}{\sigma_{\eta,i}(1-\rho_\star)^3}\sqrt{\frac{n+p}{T_i}}\right)$ and $F_i \leftarrow \frac{\widetilde{\mathcal{O}}(1)(i+1)}{1-\rho_\star}$.

8:     Set $\mathbf{K}^{(i+1)} = \boldsymbol{\Phi}_u \boldsymbol{\Phi}_x^{-1}$, where $(\boldsymbol{\Phi}_x, \boldsymbol{\Phi}_u)$ are the solution to

$$\text{minimize}_{\gamma \in [0,1)} \frac{1}{1-\gamma} \min_{\boldsymbol{\Phi}_x, \boldsymbol{\Phi}_u, V} \left\| \begin{bmatrix} Q^{1/2} & 0 \\ 0 & R^{1/2} \end{bmatrix} \begin{bmatrix} \boldsymbol{\Phi}_x \\ \boldsymbol{\Phi}_u \end{bmatrix} \right\|_{\mathcal{H}_2}$$

$$\text{s.t.} \begin{bmatrix} zI - \widehat{A}_i & -\widehat{B}_i \end{bmatrix} \begin{bmatrix} \boldsymbol{\Phi}_x \\ \boldsymbol{\Phi}_u \end{bmatrix} = I + \frac{1}{z^{F_i}}V \, , \quad \frac{\sqrt{2}\varepsilon_i}{1 - C_x\rho^{F_i+1}} \left\| \begin{bmatrix} \boldsymbol{\Phi}_\mathbf{x} \\ \boldsymbol{\Phi}_\mathbf{u} \end{bmatrix} \right\|_{\mathcal{H}_\infty} \leq \gamma \, ,$$

$$\|V\| \leq C_x\rho^{F_i+1} \, , \quad \boldsymbol{\Phi}_x \in \mathcal{S}_{F_i}(C_x, \rho) \, , \quad \boldsymbol{\Phi}_u \in \mathcal{S}_{F_i}(C_u, \rho) \, .$$

9: **end for**

---

## 3.1 Regret Upper Bounds

Our guarantees for Algorithm 1 are stated in terms of certain system specific constants, which we define here. We let $K_\star$ denote the static feedback solution to the LQR problem for $(A_\star, B_\star, Q, R)$. Next, we define $(C_\star, \rho_\star)$ such that the closed loop system $A_\star + B_\star K_\star$ belongs to $\mathcal{S}(C_\star, \rho_\star)$. Our main assumption is stated as follows.

**Assumption 3.1.** *We are given a controller $\mathbf{K}^{(0)}$ that stabilizes the true system $(A_\star, B_\star)$. Furthermore, letting $(\boldsymbol{\Phi}_x, \boldsymbol{\Phi}_u)$ denote the response of $\mathbf{K}^{(0)}$ on $(A_\star, B_\star)$, we assume that $\boldsymbol{\Phi}_x \in \mathcal{S}(C_x, \rho)$ and $\boldsymbol{\Phi}_u \in \mathcal{S}(C_u, \rho)$, where the constants $C_x, C_u, \rho$ are defined in Algorithm 1.*

The requirement of an initial stabilizing controller $\mathbf{K}^{(0)}$ is not restrictive; Dean et al. [6] provide an offline strategy for finding such a controller. Furthermore, in practice Algorithm 1 can be initialized with no controller, with random inputs applied instead to the system in the first epoch to estimate $(A_\star, B_\star)$ within an initial confidence set for which the synthesis problem becomes feasible.

Our first guarantee is on the rate of estimation of $(A_\star, B_\star)$ as the algorithm progresses through time. This result builds on recent progress [17] for estimation along trajectories of a linear dynamical system. For what follows, the notation $\widetilde{\mathcal{O}}(\cdot)$ hides absolute constants and polylog $\left(T, \frac{1}{\delta}, C_\star, \frac{1}{1-\rho_\star}, n, p, \|B_\star\|, \|K_\star\|\right)$ factors.

**Theorem 3.2.** *Fix a $\delta \in (0, 1)$ and suppose that Assumption 3.1 holds. With probability at least $1 - \delta$ the following statement holds. Suppose that $T$ is at an epoch boundary. Let $(\widehat{A}(T), \widehat{B}(T))$ denote the current estimate of $(A_\star, B_\star)$ computed by Algorithm 1 at the end of time $T$. Then, this estimate satisfies the guarantee*

$$\max\{\|\widehat{A}(T) - A_\star\|, \|\widehat{B}(T) - B_\star\|\} \leq \widetilde{\mathcal{O}}\left(\frac{C_\star\|K_\star\|}{(1-\rho_\star)^3}\frac{\sqrt{n+p}}{T^{1/3}}\right) \, .$$

Theorem 3.2 shows that Algorithm 1 achieves a consistent estimate of the true dynamics $(A_\star, B_\star)$, and learns at a rate of $\widetilde{\mathcal{O}}(T^{-1/3})$. We note that consistency of parameter estimates is not a guarantee provided by OFU or TS based approaches.

Next, we state an upper bound on the regret incurred by Algorithm 1.

**Theorem 3.3.** *Fix a $\delta \in (0,1)$ and suppose that Assumption 3.1 holds. With probability at least $1 - \delta$ the following statement holds. For all $T \geq 0$ we have that Algorithm 1 satisfies*

$$\mathsf{Regret}(T) \leq \widetilde{\mathcal{O}}\left((n+p)\frac{C_\star^4(1+\|K_\star\|)^4(1+\|B_\star\|)^2 J_\star}{(1-\rho_\star)^{16}}T^{2/3}\right).$$

*Here, the notation $\widetilde{\mathcal{O}}(\cdot)$ also hides $o(T^{2/3})$ terms.*

Our proof strategy works as follows. We first decompose regret by epochs as follows:

$$\mathsf{Regret}(T) = \sum_{i=0}^{\mathcal{O}(\log_2 T)} \sum_{k=1}^{T_i} ((x_k^{(i)})^\top Q x_k^{(i)} + (u_k^{(i)})^\top R u_k^{(i)} - J_\star),$$

where $x_k^{(i)}$ denotes the state at the $k$-th timestep in the $i$-th epoch (and similarly for $u_k^{(i)}$). By standard concentration of measure arguments, we can upper bound w.h.p. the per-epoch regret $\sum_{k=1}^{T_i}((x_k^{(i)})^\top Q x_k^{(i)} + (u_k^{(i)})^\top R u_k^{(i)} - J_\star)$ by its expected value plus a deviation term that involves the norm of $x_0^{(i)}$. Because we constrain the impulse response coefficients of the SLS response $\{\boldsymbol{\Phi}_x, \boldsymbol{\Phi}_u\}$ in Algorithm 1, this allows to easily bound $\|x_0^{(i)}\|_2$ w.h.p. again by using standard concentration arguments. We then use the SLS machinery to quantify the difference between the expected cost over the horizon $T_i$ minus $J_\star$, which yields that the regret incurred during epoch $i$ is bounded by $\widetilde{\mathcal{O}}(T_i(\sigma_{\eta,i}^2/\sigma_w^2 + \varepsilon_{i-1})J_\star)$, where $\varepsilon_{i-1}$ is the estimation error, and the $\mathcal{O}(\sigma_{\eta,i}^2/\sigma_w^2)$ contribution is the additional cost incurred from injecting exploration noise. We then bound our estimation error by $\varepsilon_i = \widetilde{\mathcal{O}}((\sigma_w/\sigma_{\eta,i})T_i^{-1/2})$ using Theorem 3.2. Setting $\sigma_{\eta,i}^2 = \sigma_w^2 T_i^{-\alpha}$, we have the per-epoch regret is bounded by $\widetilde{\mathcal{O}}(T_i^{1-\alpha} + T_i^{1-(1-\alpha)/2})$. Choosing $\alpha = 1/3$ to balance these competing powers of $T_i$ and summing over logarithmic number of epochs, we obtain a final regret of $\widetilde{\mathcal{O}}(T^{2/3})$.

The main difficulty in the proof is ensuring that the transient behavior of the resulting controllers is uniformly bounded when applied to the true system. Prior works sidestep this issue by assuming that the true dynamics lie within a (known) compact set for which the Heine-Borel theorem asserts the existence of finite constants that capture this behavior. We go a step further and work through the perturbation analysis which allows us to give a regret bound that depends only on simple quantities of the true system $(A_\star, B_\star)$. The full proof is given in the appendix.

Finally, we remark that the dependence on $1/(1-\rho_\star)$ in our results is an artifact of our perturbation analysis, and we leave sharpening this dependence to future work.

## 3.2 Regret Lower Bounds and Parameter Estimation Rates

We saw that Algorithm 1 achieves $\widetilde{\mathcal{O}}(T^{2/3})$ regret with high probability. Now we provide a matching algorithmic lower bound on the expected regret, showing that the analysis presented in Section 3.1 is sharp as a function of $T$. Moreover, our lower bound characterizes how much regret must be accrued in order to achieve a specified estimation rate for the system parameters $(A_\star, B_\star)$.

**Theorem 3.4.** *Let the initial state $x_0$ be distributed according to the steady state distribution $\mathcal{N}(0, P_\infty)$ of the optimal closed loop system, and let $\{u_t\}_{t\geq 0}$ be any sequence of inputs as in Section 2. Furthermore, let $f: \mathbb{R} \to \mathbb{R}$ be any function such that with probability $1 - \delta$ we have*

$$\lambda_{\min}\left(\sum_{k=0}^{T-1} \begin{bmatrix} x_k \\ u_k \end{bmatrix} \begin{bmatrix} x_k^\top & u_k^\top \end{bmatrix}\right) \geq f(T). \tag{3.1}$$

*Then, there exist positive values $T_0$ and $C_0$ such that for all $T \geq T_0$ we have*

$$\sum_{k=0}^{T} \mathbb{E}\left[x_k^\top Q x_k + u_k^\top R u_k - J_\star\right] \geq \frac{1}{2}(1-\delta)\lambda_{\min}(R)(1 + \sigma_{\min}(K_\star)^2)f(T-T_0) - C_0,$$

*where $T_0$ and $C_0$ are functions of $A_\star$, $B_\star$, $Q$, $R$, $\sigma_w^2$, and $n$. We note the specific form of $T_0$ and $C_0$ are given in the proof.*

The proof of the estimation error Theorem 3.2 shows that Algorithm 1 satisfies Eq. (3.1) with $f(T) = \widetilde{\mathcal{O}}(T\sigma^2_{\eta,\Theta(\log_2(T))})$. Since the exploration variance $\sigma^2_{\eta,i}$ used by Algorithm 1 during the $i$-th epoch is given by $\sigma^2_{\eta,i} = \mathcal{O}(\sigma^2_w T^{-i/3})$, we obtain the following corollary which demonstrates the sharpness of our regret analysis with respect to the scaling of $T$.

**Corollary 3.5.** *For $T > C_1(n, \delta, \sigma^2_w, A_\star, B_\star, Q, R)$ the expected regret of Algorithm 1 satisfies*

$$\sum_{k=1}^{T} \mathbb{E}\left[x_k^\top Q x_k + u_k^\top R u_k - J_\star\right] \geq \widetilde{\Omega}(\lambda_{\min}(R)(1 + \sigma_{\min}(K_\star)^2)T^{2/3}) .$$

A natural question to ask is how much regret does any algorithm accrue in order to achieve estimation error $\|\widehat{A} - A_\star\| \leq \varepsilon$ and $\|\widehat{B} - B_\star\| \leq \varepsilon$. From Theorem 3.2 we know that Algorithm 1 estimates $(A_\star, B_\star)$ at rate $\widetilde{\mathcal{O}}(T^{-1/3})$. Therefore, in order to achieve $\varepsilon$ estimation error, $T$ must be $\widetilde{\Omega}(\varepsilon^{-3})$. Hence, Theorem 3.3 implies that the regret of Algorithm 1 to achieve $\varepsilon$ estimation error is $\widetilde{\mathcal{O}}(\varepsilon^{-2})$.

Interestingly, let us consider any other Algorithm achieving $\mathcal{O}(T^\alpha)$ regret for some $0 < \alpha < 1$. Then, Theorem 3.4 suggests that the best rate achievable by such an algorithm is $\mathcal{O}(T^{-\alpha/2})$, since the minimum eigenvalue condition Eq. (3.1) governs the signal-to-noise ratio. In the case of linear-regression with independent data it is known that the minimax estimation rate is lower bounded by square root of the inverse of the minimum eigenvalue (3.1). We conjecture that the same results holds in our case. Therefore, to achieve $\varepsilon$ estimation error, any Algorithm would likely require $\Omega(\varepsilon^{-2})$ regret, showing that Algorithm 1 is optimal up to logarithmic factors in this sense. Finally, we note that while Algorithm 1 estimates $(A_\star, B_\star)$ at a rate $\widetilde{\mathcal{O}}(T^{-1/3})$, Theorem 3.4 suggests that any algorithm achieving the $\mathcal{O}(\sqrt{T})$ regret would estimate $(A_\star, B_\star)$ at a rate $\Omega(T^{-1/4})$.

## 4    Experiments

**Regret Comparison.**    We illustrate the performance of several adaptive schemes empirically. We compare the proposed robust adaptive method with non-Bayesian Thompson sampling (TS) as in Abeille and Lazaric [4] and a heuristic projected gradient descent (PGD) implementation of OFU. As a simple baseline, we use the nominal control method, which synthesizes the optimal infinite-horizon LQR controller for the estimated system and injects noise with the same schedule as the robust approach. Computational burden varies across adaptive methods due to differences in both cost and frequency of controller synthesis; implementation details and computational considerations for all methods are in Section G of the full version [7].

The comparison experiments are carried out on the following LQR problem:

$$A_\star = \begin{bmatrix} 1.01 & 0.01 & 0 \\ 0.01 & 1.01 & 0.01 \\ 0 & 0.01 & 1.01 \end{bmatrix}, \quad B_\star = I, \quad Q = 10I, \quad R = I, \quad \sigma_w = 1 . \tag{4.1}$$

This system corresponds to a marginally unstable Laplacian system where adjacent nodes are weakly connected; these dynamics were also studied by [3, 6, 18]. The cost is such that input size is penalized relatively less than state. This problem setting is amenable to robust methods due to both the cost ratio and the marginal instability, which are factors that may hurt optimistic methods. In Section H of the full version [7], we show similar results for an unstable system with large transients.

To standardize the initialization of the various adaptive methods, we use a rollout of length $T_0 = 100$ where the input is a stabilizing controller plus Gaussian noise with fixed variance $\sigma_u = 1$. This trajectory is not counted towards the regret, but the recorded states and inputs are used to initialize parameter estimates. In each experiment, the system starts from $x_0 = 0$ to reduce variance over runs. For all methods, the actual errors $\widehat{A}_t - A_\star$ and $\widehat{B}_t - B_\star$ are used rather than bounds or bootstrapped estimates. The effect of this choice on regret is small, as examined empirically in Section H of [7].

The performance of the various adaptive methods over time is compared in Figure 1. The median and 90th percentile cumulative regret over 500 instances is displayed in Figure 1a, which gives an idea of both typical and worst-case behavior. The regret of the optimal LQR controller for the true system is displayed as a baseline. Overall, the methods have very similar performance. One benefit of robustness is the guaranteed stability and bounded infinite-horizon cost at every point during

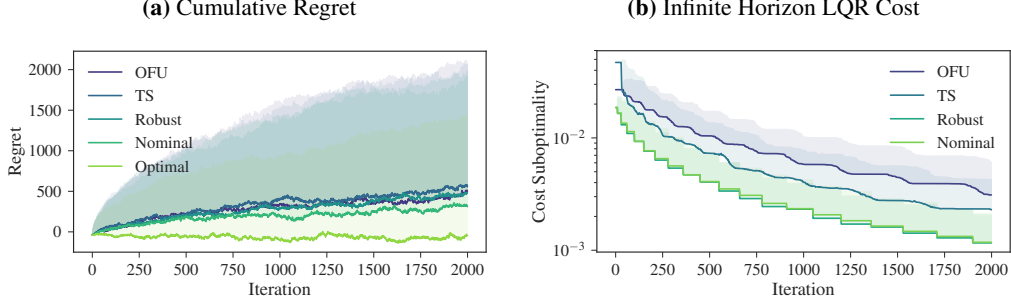

**(a)** Cumulative Regret  **(b)** Infinite Horizon LQR Cost

**Figure 1:** A comparison of different adaptive methods on 500 experiments of the marginally unstable Laplacian example in 4.1. In (a), the median and 90th percentile cumulative regret is plotted over time. In (b), the median and 90th percentile infinite-horizon LQR cost of the epoch's controller.

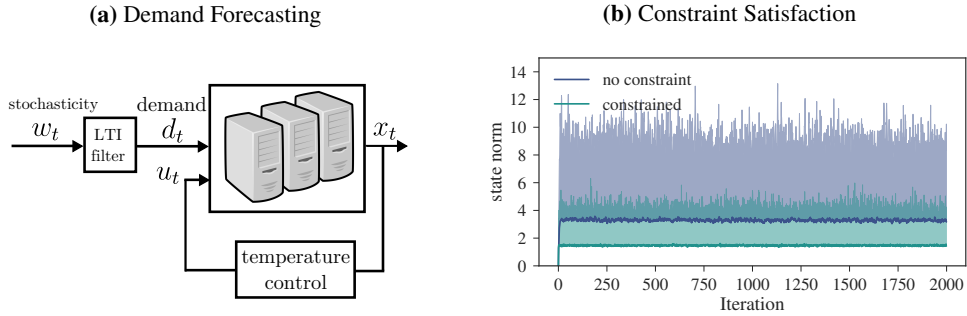

**(a)** Demand Forecasting  **(b)** Constraint Satisfaction

**Figure 2:** The addition of constraints in the robust synthesis problem can guarantee the safe execution of adaptive systems. We consider an example inspired by demand forecasting, as illustrated in (a), where the left hand side of the diagram represents unknown dynamics. The median and maximum values of $\|x_t\|_\infty$ over 500 trials are plotted for both the unconstrained and constrained synthesis problems in (b).

operation. In Figure 1b, this infinite-horizon LQR cost is plotted for the controllers played during each epoch. This value measures the cost of using each epoch's controller indefinitely, rather than continuing to update its parameters. The robust adaptive method performs relatively better than other adaptive algorithms, indicating that it is more amenable to early stopping, i.e., to turning off the adaptive component of the algorithm and playing the current controller indefinitely.

**Extension to Uncertain Environment with State Constraints.** The proposed robust adaptive method naturally generalizes beyond the standard LQR problem. We consider a disturbance forecasting example which incorporates environmental uncertainty and safety constraints. Consider a system with known dynamics driven by stochastic disturbances that are now correlated in time. We model the disturbance process as the output of an unknown autonomous LTI system, as illustrated in Figure 2a. This setting can be interpreted as a demand forecasting problem, where, for example, the system is a server farm and the disturbances represent changes in the amount of incoming jobs. If the dynamics of the correlated disturbance process are known, this knowledge can be used for more cost-effective temperature control.

We let the system $(A_\star, B_\star)$ with known dynamics be described by the graph Laplacian dynamics as in Eq. (4.1). The disturbance dynamics are unknown and are governed by a stable system transition matrix $A_d$, resulting in the following dynamics for the full system:

$$\begin{bmatrix} x_{t+1} \\ d_{t+1} \end{bmatrix} = \begin{bmatrix} A_\star & I \\ 0 & A_d \end{bmatrix} \begin{bmatrix} z_t \\ d_t \end{bmatrix} + \begin{bmatrix} B_\star \\ 0 \end{bmatrix} u_t + \begin{bmatrix} 0 \\ I \end{bmatrix} w_t , \quad A_d = \begin{bmatrix} 0.5 & 0.1 & 0 \\ 0 & 0.5 & 0.1 \\ 0 & 0 & 0.5 \end{bmatrix}.$$

The costs are set to model expensive inputs, with $Q = I$ and $R = 1 \times 10^3 I$. The controller synthesis problem in Line 8 of Algorithm 1 is modified to reflect the problem structure, and crucially, we add a constraint on the system response $\mathbf{\Phi}_x$. Further details of the formulation are explained in Section H of [7]. Figure 2b illustrates the effect. While the unconstrained synthesis results in trajectories with large state values, the constrained synthesis results in much more moderate behavior.

# 5    Conclusions and Future Work

We presented a polynomial-time algorithm for the adaptive LQR problem that provides high probability guarantees of sub-linear regret. In contrast to other approaches to this problem, our robust adaptive method guarantees stability, robust performance, and parameter estimation. We also explored the interplay between regret minimization and parameter estimation, identifying fundamental limits connecting the two.

Several questions remain to be answered. It is an open question whether a polynomial-time algorithm can achieve a regret of $\tilde{\mathcal{O}}(\sqrt{T})$. In our implementation of OFU, we observed that PGD performed quite effectively. Interesting future work is to see if the techniques of Fazel et al. [9] for policy gradient optimization on LQR can be applied to prove convergence of PGD on the OFU subroutine, which would provide an optimal polynomial-time algorithm. Moreover, we observed that OFU and TS methods in practice gave estimates of system parameters that were comparable with our method which explicitly adds excitation noise. It seems that the switching of control policies at epoch boundaries provides more excitation for system identification than is currently understood by the theory. Furthermore, practical issues that remain to be addressed include satisfying safety constraints and dealing with nonlinear dynamics; in both settings, finite-sample parameter estimation/system identification and adaptive control remain an open problem.

### Acknowledgments

We thank the anonymous reviewers for their feedback, which improved the clarity of our presentation. SD is supported by an NSF Graduate Research Fellowship under Grant No. DGE 1752814. As part of the RISE lab, HM is generally supported in part by NSF CISE Expeditions Award CCF-1730628, DHS Award HSHQDC-16-3-00083, and gifts from Alibaba, Amazon Web Services, Ant Financial, CapitalOne, Ericsson, GE, Google, Huawei, Intel, IBM, Microsoft, Scotiabank, Splunk and VMware. BR is generously supported in part by NSF award CCF-1359814, ONR awards N00014-17-1-2191, N00014-17-1-2401, and N00014-17-1-2502, the DARPA Fundamental Limits of Learning (Fun LoL) and Lagrange Programs, and an Amazon AWS AI Research Award.

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
