[Supplementary Material]

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

# A   Background on System Level Synthesis

We begin by defining two function spaces which we use extensively throughout:

$$\mathcal{RH}_\infty = \{\mathbf{M} : \mathbb{C} \longrightarrow \mathbb{C}^{n \times p} \mid \mathbf{M}(z) \text{ is rational },\ \mathbf{M}(z) \text{ is analytic on } \mathbb{D}^c \}, \qquad (\text{A.1})$$

$$\mathcal{RH}_\infty(C, \rho) = \{\mathbf{M} \in \mathcal{RH}_\infty \mid \|\mathbf{M}[k]\| \le C\rho^k ,\ k = 1, 2, ...\} . \qquad (\text{A.2})$$

Note that we use $\mathcal{S}(C, \rho)$ to denote $\mathcal{RH}_\infty(C, \rho)$ in the main body of the text.

Recall that our main object of interest is the system

$$x_{k+1} = Ax_k + Bu_k + w_k ,$$

and our goal is to design a LTI feedback control policy $\mathbf{u} = \mathbf{K}\mathbf{x}$ such that the resulting closed loop system is stable. For a given $\mathbf{K}$, we refer to the closed loop transfer functions from $\mathbf{w} \mapsto \mathbf{x}$ and $\mathbf{w} \mapsto \mathbf{u}$ as the *system response*. Symbolically, we denote these maps as $\mathbf{\Phi}_x$ and $\mathbf{\Phi}_u$. Simple algebra shows that given $\mathbf{K}$, these maps take on the form

$$\mathbf{\Phi}_x = (zI - A - B\mathbf{K})^{-1} ,\quad \mathbf{\Phi}_u = \mathbf{K}(zI - A - B\mathbf{K})^{-1} . \qquad (\text{A.3})$$

We then have the following theorem parameterizing the set of such stable closed-loop transfer functions that are achievable by a stabilizing controller $\mathbf{K}$.

**Theorem A.1** (State-Feedback Parameterization [19]). *The following are true:*

- *The affine subspace defined by*

$$[zI - A \quad -B]\begin{bmatrix}\mathbf{\Phi}_x \\ \mathbf{\Phi}_u\end{bmatrix} = I,\ \mathbf{\Phi}_x, \mathbf{\Phi}_u \in \frac{1}{z}\mathcal{RH}_\infty \qquad (\text{A.4})$$

  *parameterizes all system responses (??) from $\mathbf{w}$ to $(\mathbf{x}, \mathbf{u})$, achievable by an internally stabilizing state-feedback controller $\mathbf{K}$.*

- *For any transfer matrices $\{\mathbf{\Phi}_x, \mathbf{\Phi}_u\}$ satisfying (??), the controller $\mathbf{K} = \mathbf{\Phi}_u\mathbf{\Phi}_x^{-1}$ is internally stabilizing and achieves the desired system response (??).*

If $\mathbf{K}$ stabilizes $(A, B)$, then the LQR cost of $\mathbf{K}$ on $(A, B)$ can be written by Parseval's identity as

$$J(A, B, \mathbf{K}; \sigma_w^2 I) := \lim_{T \to \infty} \frac{1}{T}\mathbb{E}\left[\sum_{k=1}^{T} x_k^\top Q x_k + u_k^\top R u_k\right] = \sigma_w^2 \left\|\begin{bmatrix}Q^{1/2} & 0 \\ 0 & R^{1/2}\end{bmatrix}\begin{bmatrix}\mathbf{\Phi}_x \\ \mathbf{\Phi}_u\end{bmatrix}\right\|_{\mathcal{H}_2}^2 . \qquad (\text{A.5})$$

More generally, we will define $J(A, B, \mathbf{K}; \Sigma)$ to be the LQR cost when the process noise is driven by $w \overset{\text{i.i.d.}}{\sim} \mathcal{N}(0, \Sigma)$. When we omit the last argument, we mean $\sigma_w^2 = 1$, i.e. $J(A, B, \mathbf{K}) = J(A, B, \mathbf{K}; I)$.

In [6], the authors use SLS to study how uncertainty in the true parameters $(A_\star, B_\star)$ affect the LQR objective cost. Our analysis relies on these tools, which we briefly describe below.

The starting point for the theory is a characterization of all *robustly* stabilizing controllers.

**Theorem A.2** ([14]). *Suppose that the transfer matrices $\{\mathbf{\Phi}_x, \mathbf{\Phi}_u\} \in \frac{1}{z}\mathcal{RH}_\infty$ satisfy*

$$[zI - A \quad -B]\begin{bmatrix}\mathbf{\Phi}_x \\ \mathbf{\Phi}_u\end{bmatrix} = I + \mathbf{\Delta}. \qquad (\text{A.6})$$

*Then the controller $\mathbf{K} = \mathbf{\Phi}_u\mathbf{\Phi}_x^{-1}$ stabilizes the system described by $(A, B)$ if and only if $(I+\mathbf{\Delta})^{-1} \in \mathcal{RH}_\infty$. Furthermore, the resulting system response is given by*

$$\begin{bmatrix}\mathbf{x} \\ \mathbf{u}\end{bmatrix} = \begin{bmatrix}\mathbf{\Phi}_x \\ \mathbf{\Phi}_u\end{bmatrix}(I + \mathbf{\Delta})^{-1}\mathbf{w}. \qquad (\text{A.7})$$

This robustness result is used to derive a cost perturbation result for LQR.

**Lemma A.3** ([6]). *Let the controller $\mathbf{K}$ stabilize $(\widehat{A}, \widehat{B})$ and $(\boldsymbol{\Phi}_x, \boldsymbol{\Phi}_u)$ be its corresponding system response on system $(\widehat{A}, \widehat{B})$. Then if $\mathbf{K}$ stabilizes $(A, B)$, it achieves the following LQR cost*

$$\sqrt{J(A, B, \mathbf{K})} = \left\| \begin{bmatrix} Q^{\frac{1}{2}} & 0 \\ 0 & R^{\frac{1}{2}} \end{bmatrix} \begin{bmatrix} \boldsymbol{\Phi}_x \\ \boldsymbol{\Phi}_u \end{bmatrix} \left( I + \begin{bmatrix} \Delta_A & \Delta_B \end{bmatrix} \begin{bmatrix} \boldsymbol{\Phi}_x \\ \boldsymbol{\Phi}_u \end{bmatrix} \right)^{-1} \right\|_{\mathcal{H}_2}. \tag{A.8}$$

*Furthermore, letting*

$$\widehat{\boldsymbol{\Delta}} := \begin{bmatrix} \Delta_A & \Delta_B \end{bmatrix} \begin{bmatrix} \boldsymbol{\Phi}_x \\ \boldsymbol{\Phi}_u \end{bmatrix}. \tag{A.9}$$

*a sufficient condition for $\mathbf{K}$ to stabilize $(A, B)$ is that $\|\widehat{\boldsymbol{\Delta}}\|_{\mathcal{H}_\infty} < 1$. An upper bound on $\|\widehat{\boldsymbol{\Delta}}\|_{\mathcal{H}_\infty}$ is given by, for any $\alpha \in (0, 1)$,*

$$\|\widehat{\boldsymbol{\Delta}}\|_{\mathcal{H}_\infty} \leq \left\| \begin{bmatrix} \frac{\varepsilon_A}{\sqrt{\alpha}} \boldsymbol{\Phi}_x \\ \frac{\varepsilon_B}{\sqrt{1-\alpha}} \boldsymbol{\Phi}_u \end{bmatrix} \right\|_{\mathcal{H}_\infty}, \tag{A.10}$$

*where we assume that $\|A - \widehat{A}\|_2 \leq \varepsilon_A$ and $\|B - \widehat{B}\|_2 \leq \varepsilon_B$.*

# B  Synthesis Results

We first study the following infinite-dimensional synthesis problem.

$$\text{minimize}_{\gamma \in [0,1)} \frac{1}{1-\gamma} \min_{\boldsymbol{\Phi}_x, \boldsymbol{\Phi}_u} \left\| \begin{bmatrix} Q^{\frac{1}{2}} & 0 \\ 0 & R^{\frac{1}{2}} \end{bmatrix} \begin{bmatrix} \boldsymbol{\Phi}_x \\ \boldsymbol{\Phi}_u \end{bmatrix} \right\|_{\mathcal{H}_2}$$

$$\text{s.t.} \begin{bmatrix} zI - \widehat{A} & -\widehat{B} \end{bmatrix} \begin{bmatrix} \boldsymbol{\Phi}_x \\ \boldsymbol{\Phi}_u \end{bmatrix} = I, \quad \left\| \begin{bmatrix} \boldsymbol{\Phi}_x \\ \boldsymbol{\Phi}_u \end{bmatrix} \right\|_{\mathcal{H}_\infty} \leq \frac{\gamma}{\sqrt{2}\varepsilon} \tag{B.1}$$

$$\boldsymbol{\Phi}_x \in \frac{1}{z} \mathcal{R}\mathcal{H}_\infty(C_x, \rho), \ \boldsymbol{\Phi}_u \in \frac{1}{z} \mathcal{R}\mathcal{H}_\infty(C_u, \rho).$$

We will conduct our analysis assuming that this infinite-dimensional problem is solvable. Later on, we will show how to relax this problem to a finite-dimension one via FIR truncation, and show the minor modifications needed to the analysis for the guarantees to hold.

We now prove a sub-optimality guarantee on the solution to (**??**) which holds for certain choices of $\varepsilon$ and the coefficients $(C_x, \rho_x)$ and $(C_u, \rho_u)$. This result also establishes an important technical consideration, which is when the problemmmm (**??**) is feasible.

**Theorem B.1.** *Let $J_\star$ denote the minimal LQR cost achievable by any controller for the dynamical system with transition matrices $(A_\star, B_\star)$, and let $K_\star$ denote its optimal static feedback contoller. Suppose that $\mathfrak{R}_{A_\star + B_\star K_\star} \in \mathcal{R}\mathcal{H}_\infty(C_\star, \rho_\star)$ and that (wlog) $\rho_\star \geq 1/e$. Suppose furthermore that $\varepsilon$ is small enough to satisfy the following conditions:*

$$\varepsilon(1 + \|K_\star\|)\|\mathfrak{R}_{A_\star + B_\star K_\star}\|_{\mathcal{H}_\infty} \leq 1/5,$$
$$\varepsilon(1 + \|K_\star\|)C_\star \leq 1 - \rho_\star.$$

*Let $(\widehat{A}, \widehat{B})$ be any estimates of the transition matrices such that $\max\{\|\Delta_A\|, \|\Delta_B\|\} \leq \varepsilon$. Then, if $(C_x, \rho)$ and $(C_u, \rho)$ are set as,*

$$C_x = \frac{\mathcal{O}(1)C_\star}{1 - \rho_\star},$$
$$C_u = \frac{\mathcal{O}(1)\|K_\star\|C_\star}{1 - \rho_\star},$$
$$\rho = (1/4)\rho_\star + 3/4,$$

*we have that (a) the program (**??**) is feasible, (b) letting $\mathbf{K}$ denote an optimal solution to (**??**), the relative error in the LQR cost is*

$$J(A_\star, B_\star, \mathbf{K}) \leq (1 + 5\varepsilon(1 + \|K_\star\|)\|\mathfrak{R}_{A_\star + B_\star K_\star}\|_{\mathcal{H}_\infty})^2 J_\star, \tag{B.2}$$

*and (c) if furthermore* $\varepsilon(C_x + C_u) \le 2(1 - \rho_\star)$, *the response* $\{\widehat{\mathbf{\Phi}}_x, \widehat{\mathbf{\Phi}}_u\}$ *of* $\mathbf{K}$ *on the true system* $(A_\star, B_\star)$ *satisfies*

$$\widehat{\mathbf{\Phi}}_x \in \mathcal{RH}_\infty \left( \frac{\mathcal{O}(1)C_\star}{(1 - \rho_\star)^2}, 7/8 + (1/8)\rho_\star \right) ,$$

$$\widehat{\mathbf{\Phi}}_u \in \mathcal{RH}_\infty \left( \frac{\mathcal{O}(1)\|K_\star\|C_\star}{(1 - \rho_\star)^2}, 7/8 + (1/8)\rho_\star \right) .$$

*Proof.* The proof of (a) and (b) is nearly identical to that given in [6], which works by showing that $\mathbf{\Phi}_x = \mathfrak{R}_{\widehat{A}+\widehat{B}K_\star}$ and $\mathbf{\Phi}_u = K_\star \mathfrak{R}_{\widehat{A}+\widehat{B}K_\star}$ is a feasible response which gives the desired suboptimality guarantee. The only modification is that we need to find constants $C_x, C_u, \rho$ for which $\mathfrak{R}_{\widehat{A}+\widehat{B}K_\star} \in \frac{1}{z}\mathcal{RH}_\infty(C_x, \rho)$ and $K_\star \mathfrak{R}_{\widehat{A}+\widehat{B}K_\star} \in \frac{1}{z}\mathcal{RH}_\infty(C_u, \rho)$. We do this by writing

$$\mathfrak{R}_{\widehat{A}+\widehat{B}K_\star} = \mathfrak{R}_{A_\star+B_\star K_\star}(I - \mathbf{\Delta})^{-1} , \quad \mathbf{\Delta} = (\Delta_A + \Delta_B K_\star)\mathfrak{R}_{A_\star+B_\star K_\star} .$$

By the definition of $\mathbf{\Delta}$ and our assumptions, we have that

$$\mathbf{\Delta} \in \mathcal{RH}_\infty(\varepsilon(1 + \|K_\star\|)C_\star, \rho_\star) , \quad \|\mathbf{\Delta}\|_{\mathcal{H}_\infty} < 1 .$$

This places us in a position to apply Lemma **??**, from which we conclude that

$$(I - \mathbf{\Delta})^{-1} \in \mathcal{RH}_\infty \left( \mathcal{O}(1), \mathsf{Avg}(\rho_\star, 1) \right) .$$

Now applying Lemma **??** to $\mathfrak{R}_{A_\star+B_\star K_\star}(I - \mathbf{\Delta})^{-1}$, we conclude that

$$\mathfrak{R}_{\widehat{A}+\widehat{B}K_\star} \in \mathcal{RH}_\infty \left( \frac{\mathcal{O}(1)C_\star}{1 - \rho_\star}, (1/4)\rho_\star + 3/4 \right) .$$

The claims of (a) and (b) now follows.

Now for the proof of (c). Let $\{\mathbf{\Phi}_x, \mathbf{\Phi}_u\}$ be the solution to (**??**). We have that

$$\begin{bmatrix} \widehat{\mathbf{\Phi}}_x \\ \widehat{\mathbf{\Phi}}_u \end{bmatrix} = \begin{bmatrix} \mathbf{\Phi}_x \\ \mathbf{\Phi}_u \end{bmatrix} (I + \widehat{\mathbf{\Delta}})^{-1} , \quad \widehat{\mathbf{\Delta}} = [\Delta_A \quad \Delta_B] \begin{bmatrix} \mathbf{\Phi}_x \\ \mathbf{\Phi}_u \end{bmatrix} .$$

We know that $\|\widehat{\mathbf{\Delta}}\|_{\mathcal{H}_\infty} < 1$ by the constraints of the optimization problem (**??**) and furthermore,

$$\widehat{\mathbf{\Delta}} \in \mathcal{RH}_\infty(\varepsilon(C_x + C_u), \rho) .$$

By assumption we have $\varepsilon(C_x + C_u) \le 2$, from which we conclude using Lemma **??** that

$$(I + \widehat{\mathbf{\Delta}})^{-1} \in \mathcal{RH}_\infty \left( \mathcal{O}(1), \mathsf{Avg}(\rho, 1) \right) .$$

Furthermore, from Lemma **??**, we conclude that

$$\mathbf{\Phi}_x(I + \widehat{\mathbf{\Delta}})^{-1} \in \mathcal{RH}_\infty \left( \frac{C_x}{1 - \rho}, 3/4 + (1/4)\rho \right) ,$$

$$\mathbf{\Phi}_u(I + \widehat{\mathbf{\Delta}})^{-1} \in \mathcal{RH}_\infty \left( \frac{C_u}{1 - \rho}, 3/4 + (1/4)\rho \right) .$$

The claim now follows by plugging in the values of $C_x$, $C_u$, and $\rho$. $\qquad\qquad\square$

## B.1 Suboptimality bounds for FIR truncated SLS

Optimization problem (**??**) is convex but infinite dimensional, and as far as we are aware does not admit an efficient solution. In Algorithm 1, we instead propose solving the following FIR approximation to problem (**??**):

$$\text{minimize}_{\gamma \in [0,1)} \frac{1}{1 - \gamma} \min_{\mathbf{\Phi}_\mathbf{x}, \mathbf{\Phi}_\mathbf{u}, V} \left\| \begin{bmatrix} Q^{1/2} & 0 \\ 0 & R^{1/2} \end{bmatrix} \begin{bmatrix} \mathbf{\Phi}_\mathbf{x} \\ \mathbf{\Phi}_\mathbf{u} \end{bmatrix} \right\|_{\mathcal{H}_2}$$

$$\text{s.t.} \begin{bmatrix} zI - \widehat{A} & -\widehat{B} \end{bmatrix} \begin{bmatrix} \mathbf{\Phi}_\mathbf{x} \\ \mathbf{\Phi}_\mathbf{u} \end{bmatrix} = I + \frac{1}{z^F}V , \quad \frac{\sqrt{2}\varepsilon}{1 - C_x\rho^{F+1}} \left\| \begin{bmatrix} \mathbf{\Phi}_\mathbf{x} \\ \mathbf{\Phi}_\mathbf{u} \end{bmatrix} \right\|_{\mathcal{H}_\infty} \le \gamma \quad \text{(B.3)}$$

$$\|V\|_2 \le C_x\rho^{F+1} , \quad \mathbf{\Phi}_\mathbf{x} \in \frac{1}{z}\mathcal{RH}_\infty^F(C_x, \rho), \mathbf{\Phi}_\mathbf{u} \in \frac{1}{z}\mathcal{RH}_\infty^F(C_u, \rho) .$$

where here $F$ denotes the FIR truncation length used. This optimization problem can be posed as a finite dimensional semidefinite program (see Section **??**). Let $\mathbf{K}(F)$ denote the resulting controller. We begin with a lemma identifying conditions under which optimization problem (**??**) is feasible. to ease notation going forward, we let $\zeta := \varepsilon(1 + \|K_\star\|_2)\|\mathfrak{R}_{A_\star+B_\star K_\star}\|_{\mathcal{H}_\infty}$.

**Lemma B.2.** *Let the assumptions of Theorem* **??** *hold, and further assume that*

$$F_0 \geq \frac{\log(2C_x)}{\log(1/\rho)} - 1 \, .$$

*Then optimization problem* (**??**) *is feasible for any* $F \geq F_0$.

*Proof.* We construct a feasible solution as follows. Let $\mathbf{\Phi}_x = \mathfrak{R}_{\widehat{A}+\widehat{B}K_\star}(1 : F)$, $\mathbf{\Phi}_u = K_\star \mathfrak{R}_{\widehat{A}+\widehat{B}K_\star}(1 : F)$, $V = \mathfrak{R}_{\widehat{A}+\widehat{B}K_\star}(F+1)$, and $\gamma = \frac{2\sqrt{2}\zeta}{1-\zeta}$. First, the proposed $(\mathbf{\Phi}_x, \mathbf{\Phi}_u)$ are FIR of length $F$, and hence, using the same arguments as in the proof of Theorem **??**, $\mathbf{\Phi}_x \in \mathcal{RH}_\infty^F(C_x, \rho)$ and $\mathbf{\Phi}_u \in \mathcal{RH}_\infty^F(C_u, \rho)$. It then also follows immediately that $\|V\|_2 = \|\mathfrak{R}_{\widehat{A}+\widehat{B}K_\star}(F+1)\|_2 \leq C_x \rho^{F+1}$.

Note that the affine constraint

$$\begin{bmatrix} zI - \widehat{A} & -\widehat{B} \end{bmatrix} \begin{bmatrix} \mathbf{\Phi_x} \\ \mathbf{\Phi_u} \end{bmatrix} = I + \frac{1}{z^F} V \tag{B.4}$$

is equivalent to

$$\Phi_x(t+1) = \widehat{A}\Phi_x(t) + \widehat{B}\Phi_u(t), \ \Phi_x(1) = I,$$

for $1 \leq t < F$. We have by construction that the proposed $\mathbf{\Phi}_x$ and $\mathbf{\Phi}_u$ satisfy this constraint. Further, the combination of the FIR constraints and the affine constraint (**??**) impose that

$$\Phi_x(F+1) = \widehat{A}\Phi_x(F) + \widehat{B}\Phi_u(F) - V = 0.$$

Now notice that for the proposed $(\mathbf{\Phi}_x, \mathbf{\Phi}_u)$, we have that $\widehat{A}\Phi_x(F) + \widehat{B}\Phi_u(F) = (\widehat{A} + \widehat{B}K_\star)\mathfrak{R}_{\widehat{A}+\widehat{B}K_\star}(F) = \mathfrak{R}_{\widehat{A}+\widehat{B}K_\star}(F+1)$, where the last equality follows from the fact that $\mathfrak{R}_{\widehat{A}+\widehat{B}K_\star}(t+1) = (\widehat{A} + \widehat{B}K_\star)^t$. It follows that $\Phi_x(F+1) = 0$, as desired.

It remains to prove that

$$\frac{\sqrt{2}\varepsilon}{1 - C_x\rho^{F+1}} \left\| \begin{bmatrix} \mathbf{\Phi_x} \\ \mathbf{\Phi_u} \end{bmatrix} \right\|_{\mathcal{H}_\infty} \leq \frac{2\sqrt{2}\zeta}{1-\zeta} < 1.$$

The final inequality follows immediately from the assumption that $\zeta \leq 1/5$. Further, note that

$$\frac{\sqrt{2}\varepsilon}{1 - C_x\rho^{F+1}} \left\| \begin{bmatrix} \mathbf{\Phi_x} \\ \mathbf{\Phi_u} \end{bmatrix} \right\|_{\mathcal{H}_\infty} \leq 2\sqrt{2}\varepsilon \left\| \begin{bmatrix} \mathfrak{R}_{\widehat{A}+\widehat{B}K_\star} \\ K_\star \mathfrak{R}_{\widehat{A}+\widehat{B}K_\star} \end{bmatrix} \right\|_{\mathcal{H}_\infty} \leq \frac{2\sqrt{2}\zeta}{1-\zeta},$$

where the first inequality follows from the assumption on on $F_0$ and that the proposed $\mathbf{\Phi}_x$ is a truncation of $\mathfrak{R}_{\widehat{A}+\widehat{B}K_\star}$ and that the proposed $\mathbf{\Phi}_u$ is a truncation of $K_\star \mathfrak{R}_{\widehat{A}+\widehat{B}K_\star}$, and final inequality follows by applying the triangle inequality and the definition of $\zeta$. This proves the result. $\qquad\square$

Next, we use this to bound the suboptimality gap of the performance achieved by the controller implemented using the solutions of optimization problem (**??**).

**Lemma B.3.** *Let the assumptions of Lemma* **??** *hold. Fix any* $C_J > 0$, *and further let*

$$F \geq \frac{\log((1 + C_J^{-1})C_x)}{\log(1/\rho)} - 1 \, .$$

*Denote by* $(\mathbf{\Phi}_x(F), \mathbf{\Phi}_u(F), V(F), \gamma(F))$ *the optimal solution to optimization problem* (**??**), *and let* $\mathbf{K}(F) = \mathbf{\Phi}_u(F)\mathbf{\Phi}_x^{-1}(F)$. *Then*

$$J(A_\star, B_\star, \mathbf{K}(F)) \leq (1+C_J)^2(1 + \mathcal{O}(1)\varepsilon(1 + \|K_\star\|_2)\|\mathfrak{R}_{A_\star+B_\star K_\star}\|_{\mathcal{H}_\infty})^2 J_\star. \tag{B.5}$$

*Proof.* Let

$$\widehat{\mathbf{\Delta}} := \begin{bmatrix} \Delta_A & \Delta_B \end{bmatrix} \begin{bmatrix} \mathbf{\Phi}_x(F) \\ \mathbf{\Phi}_u(F) \end{bmatrix} \left( I + \frac{1}{z^F} V(F) \right)^{-1}.$$

Further note that using a similar argument to that in the proof of Lemma 4.2 of [6], one can verify that

$$\|\widehat{\mathbf{\Delta}}\|_{\mathcal{H}_\infty} \leq \frac{\sqrt{2}\varepsilon}{1 - C_x\rho^{F+1}} \left\| \begin{bmatrix} \mathbf{\Phi}_x(F) \\ \mathbf{\Phi}_u(F) \end{bmatrix} \right\|_{\mathcal{H}_\infty} \leq \gamma(F),$$

where we have exploited that $(\mathbf{\Phi}_x(F), \mathbf{\Phi}_u(F), V(F), \gamma(F))$ form a feasible solution to optimization problem (??).

Then, repeated application of Theorem **??** tells us that the performance achieved by $K(F)$ on the true system is given by

$$\sqrt{J(A_\star, B_\star, \mathbf{K}(F))} = \left\| \begin{bmatrix} Q^{\frac{1}{2}} & 0 \\ 0 & R^{\frac{1}{2}} \end{bmatrix} \begin{bmatrix} \mathbf{\Phi}_x(F) \\ \mathbf{\Phi}_u(F) \end{bmatrix} \left( I + \frac{1}{z^F} V(F) \right)^{-1} (I + \widehat{\mathbf{\Delta}})^{-1} \right\|_{\mathcal{H}_2}$$

$$\leq \frac{1}{1 - C_x \rho^{F+1}} \frac{1}{1 - \gamma(F)} \left\| \begin{bmatrix} Q^{\frac{1}{2}} & 0 \\ 0 & R^{\frac{1}{2}} \end{bmatrix} \begin{bmatrix} \mathbf{\Phi}_x(F) \\ \mathbf{\Phi}_u(F) \end{bmatrix} \right\|_{\mathcal{H}_2},$$

where the inequality follows from $\|\widehat{\mathbf{\Delta}}\|_{\mathcal{H}_\infty} \leq \gamma(F) < 1$, and $\|V(F)\|_2 \leq 1/2$ (by the assumption of $F \geq F_0$.

Denote by $(\mathbf{\Phi}_x, \mathbf{\Phi}_u, V, \gamma_0)$ the feasible solution constructed in the proof of Lemma **??**. Then,

$$\frac{1}{1 - C_x \rho^{F+1}} \frac{1}{1 - \gamma(F)} \left\| \begin{bmatrix} Q^{\frac{1}{2}} & 0 \\ 0 & R^{\frac{1}{2}} \end{bmatrix} \begin{bmatrix} \mathbf{\Phi}_x(F) \\ \mathbf{\Phi}_u(F) \end{bmatrix} \right\|_{\mathcal{H}_2} \leq \frac{1}{1 - C_x \rho^{F+1}} \frac{1}{1 - \gamma_0} \left\| \begin{bmatrix} Q^{\frac{1}{2}} & 0 \\ 0 & R^{\frac{1}{2}} \end{bmatrix} \begin{bmatrix} \mathbf{\Phi}_x \\ \mathbf{\Phi}_u \end{bmatrix} \right\|_{\mathcal{H}_2}$$

$$= \frac{1}{1 - C_x \rho^{F+1}} \frac{1}{1 - \gamma_0} \sqrt{J_F(\widehat{A}, \widehat{B}, K_\star)}$$

$$\leq \frac{1}{1 - C_x \rho^{F+1}} \frac{1}{1 - \gamma_0} \sqrt{J(\widehat{A}, \widehat{B}, K_\star)}$$

$$\leq \frac{1}{1 - C_x \rho^{F+1}} \frac{1}{1 - \gamma_0} \frac{1}{1 - \zeta} \sqrt{J_\star},$$

where the first inequality follows from the optimality of $(\mathbf{\Phi}_x(F), \mathbf{\Phi}_u(F), V(F), \gamma(F))$, the equality and second inequality from the fact that $(\mathbf{\Phi}_x, \mathbf{\Phi}_u)$ are truncations of the response of $K_\star$ on $(\widehat{A}, \widehat{B})$ to the first $F$ time steps, and the final inequality by following similar arguments to the proof of Theorem 4.1 in [6] in applying Theorem **??** and noting that

$$\left\| \begin{bmatrix} \Delta_A & \Delta_B \end{bmatrix} \begin{bmatrix} \mathfrak{R}_{\widehat{A}+\widehat{B}K_\star} \\ K_\star \mathfrak{R}_{\widehat{A}+\widehat{B}K_\star} \end{bmatrix} \right\|_{\mathcal{H}_\infty} \leq \zeta < 1.$$

We therefore have that

$$\sqrt{J(A_\star, B_\star, \mathbf{K}(F))} \leq \frac{1}{1 - C_x \rho^{F+1}} \frac{1}{1 - \gamma_0} \frac{1}{1 - \zeta} \sqrt{J_\star} \leq (1 + C_J) \frac{1}{1 - \gamma_0} \frac{1}{1 - \zeta} \sqrt{J_\star},$$

where the last inequality follows from the assumptions on $F$ stated in the Lemma. Finally, by assumption $\zeta \leq 1/5 < .8(1 + 2\sqrt{2})^{-1}$, from which it follows that $(1 - \gamma_0)^{-1}(1 - \zeta)^{-1} \leq 1 + 20\zeta$, leading to the bound

$$\sqrt{J(A_\star, B_\star, \mathbf{K}(F))} \leq (1 + C_J)(1 + 20\zeta)\sqrt{J_\star}.$$

Squaring both sides proves the result. $\qquad\qquad\square$

The following Theorem is then immediate.

**Theorem B.4.** *Let $J_\star$ denote the minimal LQR cost achievable by any controller for $(A_\star, B_\star)$. Let $K_\star$ denote the optimal controller and suppose that $\mathfrak{R}_{A_\star + B_\star K_\star} \in \mathcal{RH}_\infty(C_\star, \rho_\star)$. Fix a $C_J > 0$, and suppose that $F_0$ and $\varepsilon$ satisfy the assumptions of Lemmas **??** and **??**. Let $(\widehat{A}, \widehat{B})$ be any estimates of the transition matrices such that $\max\{\|\Delta_A\|, \|\Delta_B\|\} \leq \varepsilon$. Then, if $(C_x, \rho)$ and $(C_u, \rho)$ are set as in Lemma **??**, we have that (a) the program (??) is feasible for any truncation length $F \geq F_0$, (b) letting $\mathbf{K}(F)$ denote an optimal solution to (??) for truncation length $F$, the relative error in the LQR cost is*

$$J(A_\star, B_\star, \mathbf{K}(F)) \leq (1 + C_J)^2 (1 + \mathcal{O}(1)\varepsilon(1 + \|K_\star\|_2)\|\mathfrak{R}_{A_\star + B_\star K_\star}\|_{\mathcal{H}_\infty})^2 J_\star, \qquad (\text{B.6})$$

*and (c) if furthermore $\varepsilon(C_x + C_u) \leq \mathcal{O}(1)(1-\rho_\star)^2$, the response $\{\widehat{\mathbf{\Phi}}_x, \widehat{\mathbf{\Phi}}_u\}$ of $\mathbf{K}$ on the true system $(A_\star, B_\star)$ satisfies*

$$\widehat{\mathbf{\Phi}}_x \in \mathcal{RH}_\infty \left( \frac{\mathcal{O}(1)C_\star}{(1-\rho_\star)^3}, .999 + .001\rho_\star \right) ,$$

$$\widehat{\mathbf{\Phi}}_u \in \mathcal{RH}_\infty \left( \frac{\mathcal{O}(1)\|K_\star\|_2 C_\star}{(1-\rho_\star)^3}, .999 + .001\rho_\star \right) .$$

*Proof.* Claims (a) and (b) follow immediately from Lemmas **??** and **??**.

Now for the proof of (c). Let $\{\mathbf{\Phi}_x(F), \mathbf{\Phi}_u(F)\}$ be the solution to (**??**). Then as argued in the proof of Lemma **??**, the response achieved on the true system $(A_\star, B_\star)$ is given by

$$\begin{bmatrix} \mathbf{\Phi}_x(F) \\ \mathbf{\Phi}_u(F) \end{bmatrix} \left( I + \frac{1}{z^F}V(F) \right)^{-1} (I + \widehat{\mathbf{\Delta}})^{-1},$$

where $\widehat{\mathbf{\Delta}}$ is defined as in the proof of Lemma **??**.

We start by noting that $\mathbf{\Phi}_x(F) \in \mathcal{RH}_\infty(C_x, \rho)$, and by the assumption on $F \geq F_0$, it holds that $z^{-F}V(F) \in \mathcal{RH}_\infty(2, \rho^{1/2})$. This allows us to apply Lemma **??** to conclude that $(I + z^{-F}V(F))^{-1} \in \mathcal{RH}_\infty \left( \mathcal{O}(1)(1-\rho^{1/2})^{-1}, \mathsf{Avg}(\rho^{1/2}, 1) \right)$. Thus, applying Lemma **??** we conclude that

$$\mathbf{\Phi}_x(F) \left( I + \frac{1}{z^F}V(F) \right)^{-1} \in \mathcal{RH}_\infty \left( \frac{\mathcal{O}(1)C_x}{1-\rho^{1/2}}, \mathsf{Avg}(\mathsf{Avg}(\rho^{1/2}, 1), 1) \right).$$

A similar argument yields

$$\mathbf{\Phi}_u(F) \left( I + \frac{1}{z^F}V(F) \right)^{-1} \in \mathcal{RH}_\infty \left( \frac{\mathcal{O}(1)C_u}{1-\rho^{1/2}}, \mathsf{Avg}(\mathsf{Avg}(\rho^{1/2}, 1), 1) \right).$$

Now note that

$$\widehat{\mathbf{\Delta}} = (\Delta_A \mathbf{\Phi}_x(F) + \Delta_B \mathbf{\Phi}_u(F))(I + z^{-F}V(F))^{-1}.$$

From the previous argument, we have that

$$\Delta_A \mathbf{\Phi}_x(F)(I + z^{-F}V(F))^{-1} \in \mathcal{RH}_\infty \left( \varepsilon \frac{\mathcal{O}(1)C_x}{1-\rho^{1/2}}, \mathsf{Avg}(\mathsf{Avg}(\rho^{1/2}, 1), 1) \right),$$

$$\Delta_B \mathbf{\Phi}_u(F)(I + z^{-F}V(F))^{-1} \in \mathcal{RH}_\infty \left( \varepsilon \frac{\mathcal{O}(1)C_u}{1-\rho^{1/2}}, \mathsf{Avg}(\mathsf{Avg}(\rho^{1/2}, 1), 1) \right),$$

from which it follows that

$$\widehat{\mathbf{\Delta}} \in \mathcal{RH}_\infty \left( \varepsilon \frac{\mathcal{O}(1)(C_x + C_u)}{1-\rho^{1/2}}, \mathsf{Avg}(\mathsf{Avg}(\rho^{1/2}, 1), 1) \right).$$

By the assumptions of the Theorem, we have that $\varepsilon \frac{\mathcal{O}(1)(C_x+C_u)}{1-\rho^{1/2}} \leq 2$, allowing us to apply Lemma **??** to conclude that

$$(I + \widehat{\mathbf{\Delta}})^{-1} \in \mathcal{RH}_\infty \left( \mathcal{O}(1), \mathsf{Avg}(\mathsf{Avg}(\mathsf{Avg}(\rho^{1/2}, 1), 1), 1) \right).$$

Applying Lemma **??**, we see that

$$\mathbf{\Phi}_x(F)(I + z^{-F}V(F))^{-1}(I + \widehat{\mathbf{\Delta}})^{-1} \in \mathcal{RH}_\infty \left( \frac{\mathcal{O}(1)C_x}{1-\rho^{1/2}}, \mathsf{Avg}(\mathsf{Avg}(\mathsf{Avg}(\mathsf{Avg}(\rho^{1/2}, 1), 1), 1)1) \right)$$

$$\mathbf{\Phi}_u(F)(I + z^{-F}V(F))^{-1}(I + \widehat{\mathbf{\Delta}})^{-1} \in \mathcal{RH}_\infty \left( \frac{\mathcal{O}(1)C_u}{1-\rho^{1/2}}, \mathsf{Avg}(\mathsf{Avg}(\mathsf{Avg}(\rho^{1/2}, 1), 1), 1)1) \right)$$

Finally, to simplify these bounds to those in the Theorem statement, notice first that for $\rho \geq .4$, we have that $(1-\rho^{1/2}) > (1-\rho)^2$. Then, we also have that

$$\mathsf{Avg}(\mathsf{Avg}(\mathsf{Avg}(\mathsf{Avg}(\rho^{1/2}, 1), 1), 1)1) = \frac{31}{32} + \frac{1}{32}\rho^{1/2} = \frac{31}{32} + \frac{1}{32}(\frac{1}{4}\rho_\star + \frac{3}{4})^{1/2}.$$

Finally, one can check that for $\rho_\star \geq .4$, we have that $(\frac{1}{4}\rho_\star + \frac{3}{4})^{1/2} \leq .95 + .05\rho_\star$, leading to the bound

$$\frac{31}{32} + \frac{1}{32}(\frac{1}{4}\rho_\star + \frac{3}{4})^{1/2} \leq \frac{31.95}{32} + \frac{.05}{32}\rho_\star \leq .999 + .001\rho_\star.$$

We note that these constants are by no means optimized. $\qquad\square$

# C   Estimation

Recall that Algorithm 1 proceeds in epochs and that we denote by $x_t^{(i)}$ and $u_t^{(i)}$ the state and input at time $t$ during epoch $i$, respectively. The $i$-th epoch has length $T_i$. Note that $x_{T_i}^{(i)}$, the last state of epoch $i$, is equal to $x_0^{(i+1)}$, the first state of epoch $i + 1$.

At the end of each epoch our method estimates the parameters $(A_\star, B_\star)$ from the trajectory observed during that epoch, i.e.

$$(\widehat{A}, \widehat{B}) \in \arg\min_{A,B} \sum_{t=0}^{T_i-1} \frac{1}{2} \|x_{t+1}^{(i)} - Ax_t^{(i)} - Bu_t^{(i)}\|_2^2. \tag{C.1}$$

The goal of this section is to offer high probability confidence bounds on the estimation error of (??). For the rest of the section we suppress the dependence on the epoch index $i$ because we prove a statistical rate for a fixed epoch.

Algorithm 1 generates the inputs $u_t$ using a feedback controller $\mathbf{K}$ which stabilizes the true system $(A_\star, B_\star)$. Let $\{\mathbf{\Phi}_x, \mathbf{\Phi}_u\}$ denote the response of $\mathbf{K}$ on the true system $(A_\star, B_\star)$, and suppose that $\mathbf{\Phi}_x \in \frac{1}{z}\mathcal{R}\mathcal{H}_\infty(C_x, \rho)$ and $\mathbf{\Phi}_u \in \frac{1}{z}\mathcal{R}\mathcal{H}_\infty(C_u, \rho)$. More precisely, if $w_t \overset{\text{i.i.d.}}{\sim} \mathcal{N}(0, \sigma_w^2 I_p)$ is the process noise at time $t$ and $\eta_t \overset{\text{i.i.d.}}{\sim} \mathcal{N}(0, \sigma_\eta^2 I_p)$ is the input noise added at time $t$, then we can write

$$x_t = \Phi_x(t+1)x_0 + \sum_{k=0}^{t-1} \Phi_x(t-k)(B_\star \eta_k + w_k) \tag{C.2}$$

$$u_t = \eta_t + \Phi_u(t+1)x_0 + \sum_{k=0}^{t-1} \Phi_u(t-k)(B_\star \eta_k + w_k). \tag{C.3}$$

For the statistical analysis it is useful to consider the stochastic process $z_t = [x_t^\top, u_t^\top]^\top$. Also, we denote the filtration $\mathcal{F}_t = \sigma(x_0, \eta_0, w_0 \ldots, \eta_{t-1}, w_{t-1}, \eta_t)$. It is clear that the process $\{z_t\}_{t \geq 0}$ is $\{\mathcal{F}_t\}_{t \geq 0}$-adapted. Throughout this section we assume that $C_u, C_x \geq 1$ and denote $C_K^2 := nC_x^2 + p\bar{C}_u^2$.

## C.1   Estimation after one epoch

Throughout this section we assume that $\sigma_\eta \leq \sigma_w$. This condition is not needed for achieving the necessary statistical rate of estimation of $(A, B)$, but it aids in simplifying several algebraic quantities.

**Proposition C.1.** *Let $x_0 \in \mathbb{R}^n$ be any initial state, let $\sigma_\eta \leq \sigma_w$, and assume that a trajectory $\{(x_t, u_t)\}_{t=0}^{T-1}$ is observed. Furthermore, suppose the inputs $u_t \in \mathbb{R}^p$ are generated by a feedback controller $\mathbf{K}$ which stabilizes and achieves a response $\{\mathbf{\Phi}_x, \mathbf{\Phi}_u\}$ on $(A_\star, B_\star)$ with $\mathbf{\Phi}_x \in \frac{1}{z}\mathcal{R}\mathcal{H}_\infty(C_x, \rho)$ and $\mathbf{\Phi}_u \in \frac{1}{z}\mathcal{R}\mathcal{H}_\infty(C_u, \rho)$. Then, the error of the OLS estimator $(\widehat{A}, \widehat{B})$ from Eq. ?? satisfies with probability $1 - \delta$ the guarantee*

$$\max\left\{\begin{matrix}\|\widehat{A}-A_\star\|, \\ \|\widehat{B}-B_\star\|\end{matrix}\right\} \lesssim \frac{\sigma_w C_u}{\sigma_\eta} \sqrt{\frac{(n+p)}{T}\log\left(1 + \frac{pC_u}{\delta} + \frac{\sigma_w}{\sigma_\eta}\frac{\rho C_u C_K}{\delta(1-\rho^2)}\left(1 + \|B_\star\| + \frac{\|x_0\|_2}{\sigma_w\sqrt{T}}\right)\right)},$$

*as long as*

$$T \gtrsim (n+p)\log\left(1 + \frac{pC_u^2}{\delta} + \frac{\sigma_w^2}{\sigma_\eta^2}\frac{\rho^2 C_u^2 C_K^2}{\delta(1-\rho^2)}\left(1 + \|B_\star\|^2 + \frac{\|x_0\|_2^2}{\sigma_w^2 T}\right)\right). \tag{C.4}$$

The proof of this result follows from a result by Simchowitz et al. [17] on the estimation of linear response time-series. We present that result in the context of our problem. Let $M_\star = [A_\star, B_\star]$, and recall that $z_t = [x_t^\top, y_t^\top]^\top$. Then, the OLS estimator (??) can be written in the form

$$\widehat{M} \in \arg\min_M \sum_{t=0}^{T-1} \frac{1}{2}\|x_{t+1} - Mz_t\|_2^2. \tag{C.5}$$

The process $\{z_t\}_{t\geq 0}$ is said to satisfy the $(k,\nu,\beta)$-*block martingale small-ball* (BMSB) condition if for any $j \geq 0$ and $v \in \mathbb{R}^{n+p}$, one has that

$$\frac{1}{k}\sum_{i=1}^{k} \mathbb{P}\left(|\langle v, z_{j+i}\rangle| \geq \nu\right) \geq \beta \quad \text{almost surely.}$$

This condition is used for characterizing the size of the minimum eigenvalue of the covariance matrix $\sum_{t=0}^{T-1} z_t z_t^\top$. A larger $\nu$ guarantees a larger lower bound of the minimum eigenvalue. In the context of our problem the result by Simchowitz et al. [17] translates as follows.

**Theorem C.2** (Simchowitz et al. [17]). *Fix $\epsilon, \delta \in (0,1)$. For every $T$, $k$, $\nu$, and $\beta$ such that $\{z_t\}_{t\geq 0}$ satisfies the $(k,\nu,\beta)$-BMSB and*

$$\left\lfloor \frac{T}{k} \right\rfloor \gtrsim \frac{n+p}{\beta^2} \log\left(1 + \frac{\sum_{t=0}^{T-1} \mathbf{Tr}(\mathbb{E}z_t z_t^\top)}{k\lfloor T/k\rfloor \beta^2 \nu^2 \delta}\right),$$

*the estimate $\widehat{M}$ defined in Eq. ?? satisfies the following statistical rate*

$$\mathbb{P}\left(\|\widehat{M} - M\|_2 > \frac{\mathcal{O}(1)\sigma_w}{\beta\nu}\sqrt{\frac{n+p}{k\lfloor T/k\rfloor}\log\left(1 + \frac{\sum_{t=0}^{T-1} \mathbf{Tr}(\mathbb{E}z_t z_t^\top)}{k\lfloor T/k\rfloor \beta^2 \nu^2 \delta}\right)}\right) \leq \delta.$$

Therefore, in order to apply this result we need to find $k$, $\nu$, and $\beta$ such that $\{z_t\}_{t\geq 0}$ satisfies the $(k,\nu,\beta)$-BMSB condition, and we also have to upper bound the trace of the covariance of $z_t$. The next two lemmas address these two issues.

**Lemma C.3.** *Let $x_0$ be any initial state in $\mathbb{R}^n$ and let $\{u_t\}_{t\geq 0}$ be the sequence of inputs generated according to (??), and assume $\sigma_\eta \leq \sigma_w$. Then, the process $z_t = [x_t^\top, u_t^\top]^\top$ satisfies the*

$$\left(1, \frac{\sigma_\eta}{2C_u}, \frac{3}{20}\right) \text{ BMSB condition.}$$

*Proof.* For all $t \geq 1$, denote

$$\xi_t = u_t - \eta_t - \Phi_u(1)w_{t-1}$$

$$= \Phi_u(t+1)x_0 + \sum_{k=0}^{t-2} \Phi_u(t-k)(B_\star \eta_k + w_k) + \Phi_u(1)B_\star \eta_{t-1}.$$

Therefore, we have

$$\begin{bmatrix} x_{t+1} \\ u_{t+1} \end{bmatrix} = \begin{bmatrix} A_\star x_t + B_\star u_t \\ \xi_{t+1} \end{bmatrix} + \begin{bmatrix} I_n & 0 \\ \Phi_u(1) & I_p \end{bmatrix}\begin{bmatrix} w_t \\ \eta_{t+1} \end{bmatrix},$$

and hence

$$\begin{bmatrix} x_{t+1} \\ u_{t+1} \end{bmatrix} \Big| \mathcal{F}_t \sim \mathcal{N}\left(\begin{bmatrix} A_\star x_t + B_\star u_t \\ \xi_{t+1} \end{bmatrix}, \begin{bmatrix} \sigma_w^2 I_n & \sigma_w^2 \Phi_u(1)^\top \\ \sigma_w^2 \Phi_u(1) & \sigma_w^2 \Phi_u(1)\Phi_u(1)^\top + \sigma_\eta^2 I_p \end{bmatrix}\right).$$

Denote by $\mu_{z,t}$ and $\Sigma_z$ the mean and covariance of this multivariate normal distribution. Recall that we denote $z_t = [x_t^\top, u_t^\top]^\top$. Let $v \in \mathbb{R}^{n+p}$ and then $\langle v, z_t\rangle \sim \mathcal{N}(\langle v, \mu_{z,t}\rangle, v^\top \Sigma_z v)$. Therefore,

$$\mathbb{P}\left(|\langle v, z_t\rangle| \geq \sqrt{\lambda_{\min}(\Sigma_z)}\right) \geq \mathbb{P}\left(|\langle v, z_t\rangle| \geq \sqrt{v^\top \Sigma_z v}\right)$$

$$\geq \mathbb{P}\left(|\langle v, z_t - \mu_{z,t}\rangle| \geq \sqrt{v^\top \Sigma_z v}\right) \geq 3/10,$$

where the last two inequalities follow because for any $\mu, \sigma^2 \in \mathbb{R}$ and $\omega \sim \mathcal{N}(0, \sigma^2)$ we have

$$\mathbb{P}(|\mu + \omega| \geq \sigma) \geq \mathbb{P}(|\omega| \geq \sigma) \geq 3/10.$$

Since $\Phi_u \in \frac{1}{z}\mathcal{R}\mathcal{H}_\infty(C_u, \rho)$ we have $\|\Phi_u(1)\| \leq C_u$. Then, by a simple argument based on a Schur complement (detailed in Lemma ??) it follows that

$$\lambda_{\min}(\Sigma_z) \geq \sigma_\eta^2 \min\left(\frac{1}{2}, \frac{\sigma_w^2}{2\sigma_w^2 C_u^2 + \sigma_\eta^2}\right).$$

The conclusion follows since $C_u \geq 1$. $\qquad\square$

**Lemma C.4.** *Let $\sigma_\eta \leq \sigma_w$. Then, the process $z_t = [x_t^\top, u_t^\top]^\top$ satisfies*

$$\sum_{t=0}^{T-1} \mathbf{Tr}\left(\mathbb{E}z_t z_t^\top\right) \leq \sigma_\eta^2 pT + \sigma_w^2 \frac{\rho^2 C_K^2 T}{(1-\rho^2)}\left(1 + \|B_\star\|^2 + \frac{\|x_0\|_2^2}{\sigma_w^2 T}\right).$$

*Proof.* Now, note that

$$\mathbb{E}z_t z_t^\top = \begin{bmatrix} \Phi_x(t+1) \\ \Phi_u(t+1) \end{bmatrix} x_0 x_0^\top \begin{bmatrix} \Phi_x(t+1) \\ \Phi_u(t+1) \end{bmatrix}^\top + \begin{bmatrix} 0 & 0 \\ 0 & \sigma_\eta^2 I_p \end{bmatrix}$$
$$+ \sum_{k=0}^{t-1} \begin{bmatrix} \Phi_x(t-k) \\ \Phi_u(t-k) \end{bmatrix} (\sigma_\eta^2 B_\star B_\star^\top + \sigma_w^2 I_n) \begin{bmatrix} \Phi_x(t-k) \\ \Phi_u(t-k) \end{bmatrix}^\top.$$

Since for all $j \geq 1$ we have $\|\Phi_x(j)\| \leq C_x \rho^j$ and $\|\Phi_u(j)\| \leq C_u \rho^j$, we obtain

$$\mathbf{Tr}\,\mathbb{E}z_t z_t^\top \leq p\sigma_\eta^2 + (nC_x^2 + pC_u^2)\left(\rho^{2t+2}\|x_0\|_2^2 + (\sigma_w^2 + \sigma_\eta^2\|B_\star\|^2)\sum_{k=1}^{t} \rho^{2k}\right)$$

Therefore, we get that

$$\sum_{t=0}^{T-1} \mathbf{Tr}\,\mathbb{E}z_t z_t^\top \leq p\sigma_\eta^2 T + \frac{\rho^2 T}{1-\rho^2}(nC_x^2 + pC_u^2)(\sigma_w^2 + \sigma_\eta^2\|B_\star\|^2) + \frac{\rho^2}{1-\rho^2}(nC_x^2 + pC_u^2)\|x_0\|_2^2,$$

and the conclusion follows by simple algebra. $\qquad\square$

Proposition **??** follows from Theorem **??**, Lemma **??**, Lemma **??**, and simple algebra.

## C.2 Stitching the epochs together

We start by bounding with high probability the size of the initial states of the epochs. Recall that epoch $i$ has length $T_i$ and that we denote by $x_{T_i}^{(i)}$ the last state of the epoch $i$, which is equal to the first state $x_0^{(i+1)}$ of the epoch $i + 1$. For simplicity we assume that $x_0^{(0)} = 0$, an assumption that is not restrictive in any way.

**Lemma C.5.** *Fix $\delta \in (0, 1)$, $r > 0$, and an epoch $i$. Assume that for all $k \leq i$ the epoch length $T_k$ is large enough so that $C_x \rho^{T_k} \leq \rho^r$. Then, for any $t \geq 0$ we have*

$$\mathbb{P}\left(\|x_0^{(i+1)}\|_2 \geq \sigma_w(\sqrt{n} + t)\frac{C_x \rho(1 + \|B_\star\|)}{(1-\rho^r)(1-\rho^2)}\right) \leq \exp\left(-\frac{t^2}{2}\right).$$

*Proof.* From Eq. (**??**) we have that

$$x_0^{(i+1)} = \Phi_x^{(i)}(T_i + 1)x_0^{(i)} + \underbrace{\sum_{j=0}^{T_i-1} \Phi_x^{(i)}(T_i - 1 - j)(B_\star \eta_j^{(i)} + w_j^{(i)})}_{\xi_i},$$

where we denoted the sum over disturbances during the epoch $i$ by $\xi_i$. Therefore,

$$\|x_0^{(i+1)}\|_2 \leq C_x \rho^{T_i}\|x_0^{(i)}\|_2 + \|\xi_i\|_2$$
$$\leq \rho^r\|x_0^{(i)}\|_2 + \|\xi_i\|_2$$
$$\leq \sum_{k=0}^{i} \rho^{r(i-k)}\|\xi_k\|_2.$$

By definition $\xi_k$ is a zero-mean multivariate Gaussian random vector with covariance

$$\Sigma_{x,k} := \sum_{j=0}^{T_k-1} \Phi_x^{(k)}(T_k - 1 - j)(\sigma_w^2 + \sigma_{\eta,k}^2 B_\star B_\star^\top)\Phi_x^{(k)}(T_k - 1 - j)^\top,$$

whose top eigenvalue is upper bounded by

$$\sum_{j=0}^{T_k-2} C_x^2(\sigma_w^2 + \sigma_{\eta,k}^2\|B_\star\|^2)\rho^{2(T_k-1-j)} \leq (\sigma_w^2 + \sigma_{\eta,k}^2\|B_\star\|^2)\frac{C_x^2\rho^2}{1-\rho^2}$$

$$\leq \sigma_w^2(1+\|B_\star\|^2)\frac{C_x^2\rho^2}{1-\rho^2}, \tag{C.6}$$

where the last inequality follows because $\sigma_{\eta,k} \leq \sigma_w$.

Then, we can write $\|\xi_k\|_2$ as $\|\Sigma_{x,k}^{1/2}\omega_k\|_2$, where $\omega_k$ is a standard Gaussian random vector distributed according to $\mathcal{N}(0, I_n)$, and hence $\|\xi_i\|_2$ is a Lipschitz function of $\omega_i$ with Lipschitz constant equal to squared root of (**??**). Hence, $\|x_0^{(i)}\|_2$ is a Lipschitz function of standard normal random variables with the Lipschitz constant

$$\sqrt{\sigma_w^2\frac{(1+\|B_\star\|^2)}{1-\rho^r}\frac{C_x^2\rho^2}{1-\rho^2}}.$$

By the concentration of Lipschitz functions of isotropic Gaussians, for $\nu \geq 0$, we have that

$$\mathbb{P}\left(\|x_0^{(i+1)}\|_2 \geq \mathbb{E}\|x_0^{(i+1)}\|_2 + \nu\right) \leq \exp\left(-\frac{\nu^2(1-\rho^2)(1-\rho^r)}{2\sigma_w^2\rho^2(1+\|B_\star\|^2)C_x}\right).$$

By Jensen's inequality we have that

$$\mathbb{E}\|x_0^{(i+1)}\|_2 \leq \sqrt{\mathbb{E}\|x_0^{(i+1)}\|_2^2} \leq \sqrt{\sum_{k=0}^{i}\rho^{r(i-k)}\,\mathbf{Tr}\left(\mathbb{E}\xi_k\xi_k^\top\right)}$$

$$\leq \sqrt{n\sigma_w^2\frac{(1+\|B_\star\|^2)}{1-\rho^r}\frac{C_x^2\rho^2}{1-\rho^2}}.$$

The conclusion follows. $\qquad\square$

We are now ready to prove that the statistical rate holds across epochs. In order to achieve this, we need the statistical rate after the first epoch to be small enough to satisfy the feasibility constraints on $\varepsilon$ given in Theorem **??** for the IIR case and given in Theorem **??** for the FIR truncated case. Once this occurs, we immediately have feasibility at the next epoch (w.h.p.), and iterating the argument gives us recursive feasibility (w.h.p.).

**Theorem C.6.** *Fix a $\delta \in (0,1)$. For the IIR case, let $C_x, C_u, \rho$ be defined as*

$$C_x = \frac{\mathcal{O}(1)C_\star}{(1-\rho_\star)^2},$$

$$C_u = \frac{\mathcal{O}(1)\|K_\star\|C_\star}{(1-\rho_\star)^2},$$

$$\rho = (1/8)\rho_\star + (7/8),$$

*and for the FIR case, let $C_x, C_u, \rho$ be defined as*

$$C_x = \frac{\mathcal{O}(1)C_\star}{(1-\rho_\star)^3},$$

$$C_u = \frac{\mathcal{O}(1)\|K_\star\|C_\star}{(1-\rho_\star)^3},$$

$$\rho = 0.001\rho_\star + .999,$$

*where $(C_\star, \rho_\star)$ are as defined in Theorem **??** (resp. Theorem **??**), and suppose (wlog) that $C_x \geq 1$ and $C_u \geq 1$. Let the length of epoch $i \in \{0, 1, 2, ...\}$ be $T_i = C_T 2^i$ time steps and let the injected*

*noise variance at epoch $i$ be $\sigma_{\eta,i}^2 = \sigma_w^2 2^{-i/3}$. Suppose the constant $C_T$ is large enough to satisfy the following inequalities,*

$$C_T \geq \frac{\log(2C_x)}{\log(1/\rho)}, \tag{C.7}$$

$$C_T \gtrsim \frac{1}{2^i}\left(n + \log\left(\frac{i+1}{\delta}\right)\right) \; \text{for all } i = 0, 1, 2, \dots, \tag{C.8}$$

$$C_T \gtrsim \frac{(n+p)}{2^i}\log\left(1 + (i+1)^2\frac{pC_u^2}{\delta} + (i+1)^2 2^{i/3}\frac{\rho^2 C_u^2 C_K^2}{\delta(1-\rho^2)}\left(\frac{C_x^2(1+\|B_\star\|)^2}{(1-\rho)^2}\right)\right) \tag{C.9}$$

$$\text{for all } i = 0, 1, 2, \dots,$$

$$C_T \gtrsim \frac{(n+p)}{2^{2i/3}}\frac{C_u^2(C_x+C_u)^2}{(1-\rho_\star)^\alpha} \tag{C.10}$$

$$\times \log\left(1 + (i+1)\frac{pC_u}{\delta} + (i+1)2^{i/6}\frac{\rho C_u C_K}{\delta(1-\rho^2)}\left(\frac{C_x(1+\|B_\star\|)}{1-\rho}\right)\right)$$

$$\text{for all } i = 0, 1, 2, \dots,$$

*where above $\alpha = 2$ for the IIR case and $\alpha = 4$ for the FIR case. Then, with probability $1 - \delta$, the following two statements hold. First, for all epochs $i$, the norm of the first state at the beginning of each epoch satisfies*

$$\|x_0^{(i)}\|_2 \lesssim \sigma_w\left(\sqrt{n} + \sqrt{\log\left(\frac{i+1}{\delta}\right)}\right)\frac{C_x\rho(1+\|B_\star\|)}{1-\rho^2}. \tag{C.11}$$

*Second, for all epochs $i$, the OLS estimate $(\widehat{A}^{(i)}, \widehat{B}^{(i)})$ satisfies the statistical rate*

$$\max\left\{\begin{array}{c}\|\widehat{A}^{(i)}-A\|, \\ \|\widehat{B}^{(i)}-B\|\end{array}\right\} \lesssim \frac{\sigma_w C_u}{\sigma_{\eta,i}}\sqrt{\frac{(n+p)}{T_i}\log\left(1 + (i+1)\frac{pC_u}{\delta} + (i+1)\frac{\sigma_w}{\sigma_{\eta,i}}\frac{\rho C_u C_K}{\delta(1-\rho^2)}\left(\frac{C_x(1+\|B_\star\|)}{1-\rho}\right)\right)}. \tag{C.12}$$

*Proof.* For this proof, we set $r = \log(2)/\log(1/\rho)$.

By Theorem **??** for the IIR case and Theorem **??** for the FIR case, we know that the true responses $\{\mathbf{\Phi}_x, \mathbf{\Phi}_u\}$ of the synthesized controllers $\mathbf{K}_i$ on $(A_\star, B_\star)$ at every epoch satisfy $\mathbf{\Phi}_x \in \mathcal{RH}_\infty(C_x, \rho)$ and $\mathbf{\Phi}_u \in \mathcal{RH}_\infty(C_u, \rho)$.

Because of the assumption (**??**) on $C_T$ we have $C_x\rho^{T_i} \leq \rho^r$. Therefore, we can apply Lemma **??** with $t^2 = \log(\mathcal{O}(1)(i+1)^2/\delta)$ to obtain that with probability at least $1 - \delta/2$ the norm of $x_0^{(i)}$ for all epochs $i$ satisfies

$$\|x_0^{(i)}\|_2 \lesssim \sigma_w\left(\sqrt{n} + \sqrt{\log\left(\frac{i+1}{\delta}\right)}\right)\frac{C_x\rho(1+\|B_\star\|)}{(1-\rho^r)(1-\rho^2)}.$$

Furthermore, by the assumption (**??**) on $C_T$ we have that with probability at least $1 - \delta/2$,

$$\frac{\|x_0^{(i)}\|_2^2}{\sigma_w^2 T_i} \leq \frac{C_x^2(1+\|B_\star\|)^2}{(1-\rho)^2}.$$

Our assumption (**??**) means that condition (**??**) is satisfied for each epoch $i$ and therefore under the assumption the SLS program is feasible at every iteration, we can invoke Proposition **??** with $\delta = \mathcal{O}(1)\delta/(i+1)^2$ and reach the desired conclusions.

To show feasibility of the SLS at every epoch, Theorem **??** for the IIR case requires that

$$\varepsilon(i) \leq \mathcal{O}(1)\frac{1-\rho_\star}{C_x + C_u},$$

and Theorem **??** for the FIR case requires that

$$\varepsilon(i) \leq \mathcal{O}(1) \frac{(1 - \rho_\star)^2}{C_x + C_u},$$

where $\varepsilon(i)$ is our statistical upper bound on the errors $\max \left\{ \begin{smallmatrix} \|\widehat{A}^{(i)} - A\|, \\ \|\widehat{B}^{(i)} - B\| \end{smallmatrix} \right\}$. This condition is ensured by our assumption (**??**) on $C_T$. $\square$

We now remark on the satisfiability of the constraints on $C_T$ given by (**??**), (**??**), and (**??**). For (**??**) and (**??**) (resp. (**??**)), the RHS grows like $\text{poly}(i)/2^i$ (resp. $\text{poly}(i)/2^{2i/3}$) and hence the supremum of the RHS (as a function of $i$) is achieved for some finite $i$. Therefore, we have that $C_T$ satisfies in the IIR case

$$C_T = \widetilde{\mathcal{O}} \left( \max \left\{ \frac{1}{1 - \rho_\star}, n, (n + p) \frac{C_\star^4 (1 + \|K_\star\|)^4}{(1 - \rho_\star)^8} \right\} \right)$$

$$= \widetilde{\mathcal{O}} \left( (n + p) \frac{C_\star^4 (1 + \|K_\star\|)^4}{(1 - \rho_\star)^8} \right), \tag{C.13}$$

and that $C_T$ satisfies in the FIR case

$$C_T = \widetilde{\mathcal{O}} \left( \max \left\{ \frac{1}{1 - \rho_\star}, n, (n + p) \frac{C_\star^4 (1 + \|K_\star\|)^4}{(1 - \rho_\star)^{10}} \right\} \right)$$

$$= \widetilde{\mathcal{O}} \left( (n + p) \frac{C_\star^4 (1 + \|K_\star\|)^4}{(1 - \rho_\star)^{16}} \right). \tag{C.14}$$

# D   Regret Decomposition and Analysis

We use the following regret decomposition, and for simplicity we assume that $T$ is such that $T_0 + T_1 + ... + T_{E-1} = T$ for some $E$. Note that $E = O(\log_2 T)$.

$$\text{Regret}(T) = \sum_{k=1}^{T} (x_k^\top Q x_k + u_k^\top R u_k - J_\star) = \sum_{i=0}^{E-1} \sum_{j=1}^{T_i} (x_{i,j}^\top Q x_{i,j} + u_{i,j}^\top R u_{i,j} - J_\star). \tag{D.1}$$

Here, we let $x_{i,j}$ denote the $j$-th state at the $i$-th epoch (and similarly for $u_{i,j}$). Our definition of regret is defined for a given realization, as opposed to in expectation. However, in our analysis so far we have considered sub-optimality guarantees in expectation. Hence, our first concern is going from a realization to expectation.

Denote by $J_T(A, B, \mathbf{K}; \Sigma)$ the expected cost incurred by a (stabilizing) feedback policy $\mathbf{K}$ over a finite horizon $T$ on system $(A, B)$ being driven by process noise $w \overset{\text{i.i.d.}}{\sim} \mathcal{N}(0, \Sigma)$ and starting from an initial condition of $x_0 = 0$, i.e.,

$$J_T(A, B, \mathbf{K}; \Sigma) := \sum_{k=1}^{T} \mathbb{E} \left[ x_k^\top Q x_k + u_k^\top R u_k \right]. \tag{D.2}$$

Recall also that $J(A, B, \mathbf{K}; \Sigma)$ is the infinite-horizon LQR cost of $\mathbf{K}$ in feedback with $(A, B)$. We now state some basic properties of $J_T$ and $J$. We omit the proofs of these properties as they are standard.

**Lemma D.1.** *The following are true*

(i) $J_T(A, B, \mathbf{K}; \Sigma) \leq T J(A, B, \mathbf{K}; \Sigma)$,

(ii) $J(A, B, \mathbf{K}; \Sigma_1 + \Sigma_2) = J(A, B, \mathbf{K}; \Sigma_1) + J(A, B, \mathbf{K}; \Sigma_2)$,

(iii) $J(A, B, \mathbf{K}; \alpha\Sigma) = \alpha J(A, B, \mathbf{K}; \Sigma)$ *for* $\alpha > 0$,

(iv) $J(A, B, \mathbf{K}; \Sigma_1) \leq J(A, B, \mathbf{K}; \Sigma_2)$ *if* $\Sigma_1 \preceq \Sigma_2$.

From these properties, we immediately conclude that

$$J_T(A, B, \mathbf{K}; \sigma_w^2 I + \sigma_\eta^2 B B^\top) \leq T \left( 1 + \frac{\sigma_\eta^2 \|B\|^2}{\sigma_w^2} \right) J(A, B, \mathbf{K}; \sigma_w^2 I), \qquad (\text{D.3})$$

a fact we will make use of later on.

The following lemma relates the finite horizon cost to its expectation.

**Lemma D.2.** *Let $\mathbf{K}$ be a feedback policy that stabilizes $(A, B)$ and that induces system responses $\Phi_x \in \mathcal{RH}_\infty(C_x, \rho)$ and $\Phi_u \in \mathcal{RH}_\infty(C_u, \rho)$. Suppose that the system $(A, B)$ is started at $x_0 = x$ and is driven by process noise $w \overset{i.i.d.}{\sim} \mathcal{N}(0, \Sigma)$ with $\Sigma \succ 0$ and $\|\Sigma\| \leq \sigma^2$. Then with probability at least $1 - \frac{1}{\delta}$ over the randomness of the process noise,*

$$\sum_{k=1}^{T} x_k^\top Q x_k + u_k^\top R u_k \leq J_T(A, B, \mathbf{K}; \Sigma) + C_c \cdot \mathcal{O}\left( \|x\|_2^2 + \sigma^2 (\sqrt{nT \log\left(\frac{2}{\delta}\right)} + \log\left(\frac{2}{\delta}\right)) \right),$$

(D.4)

*for $C_c := (1 - \rho)^{-2} (\|Q\| C_x^2 + \|R\| C_u^2)$.*

*Proof.* Writing $\Phi_x$ as $\Phi_x = \sum_{k=1}^{\infty} \Phi_x(k) z^{-k}$, we define the following finite-horizon truncations of its block-Toeplitz representation:

$$\Phi_{x,T} := \begin{bmatrix} \Phi_x(1) & & \\ \vdots & \ddots & \\ \Phi_x(T) & \cdots & \Phi_x(1) \end{bmatrix} \qquad \Phi_{x,+} := \begin{bmatrix} \Phi_x(2) \\ \Phi_x(3) \\ \vdots \\ \Phi_x(T+1) \end{bmatrix}.$$

We let $\Phi_{u,T}$ and $\Phi_{u,T,+}$ define similar matrices for $\Phi_u$. Using these definitions, we can write

$$\sum_{k=1}^{T} x_k^\top Q x_k + u_k^\top R u_k = \begin{bmatrix} x \\ \omega \end{bmatrix}^\top \begin{bmatrix} M_{11} & M_{12} \\ M_{12}^\top & M_{22} \end{bmatrix} \begin{bmatrix} x \\ \omega \end{bmatrix},$$

for

$$\omega^\top = \begin{bmatrix} w_0^\top & w_1^\top & \cdots & w_{T-1}^\top \end{bmatrix}$$

$$M_{11} = \begin{bmatrix} \Phi_{x,+} \\ \Phi_{u,+} \end{bmatrix}^\top \begin{bmatrix} \mathcal{Q} & \\ & \mathcal{R} \end{bmatrix} \begin{bmatrix} \Phi_{x,+} \\ \Phi_{u,+} \end{bmatrix}$$

$$M_{12} = \begin{bmatrix} \Phi_{x,+} \\ \Phi_{u,+} \end{bmatrix}^\top \begin{bmatrix} \mathcal{Q} & \\ & \mathcal{R} \end{bmatrix} \begin{bmatrix} \Phi_{x,T} \\ \Phi_{u,T} \end{bmatrix}$$

$$M_{22} = \begin{bmatrix} \Phi_{x,T} \\ \Phi_{u,T} \end{bmatrix}^\top \begin{bmatrix} \mathcal{Q} & \\ & \mathcal{R} \end{bmatrix} \begin{bmatrix} \Phi_{x,T} \\ \Phi_{u,T} \end{bmatrix},$$

where $\mathcal{Q} := \text{blkdiag}(Q)$ and $\mathcal{R} := \text{blkdiag}(R)$ are block-diagonal matrices of compatible dimension. With these definitions, one can then check that $\mathbf{Tr}\, M_{22} \text{blkdiag}(\Sigma) = J_T(A, B, \mathbf{K}; \Sigma)$.

Finally, given that $\Phi_{x,+}, \Phi_{x,T}$ are sub-matrices of the block-Toeplitz representation of $\Phi_x$, it follows that $\max\{\|\Phi_{x,+}\|, \|\Phi_{x,T}\|\} \leq \|\Phi_x\|_{\mathcal{H}_\infty} \leq \frac{C_x}{1-\rho}$, where the last inequality follows from Lemma **??**. Similarly, we have that $\max\{\|\Phi_{u,+}\|, \|\Phi_{u,T}\|\} \leq \|\Phi_u\|_{\mathcal{H}_\infty} \leq \frac{C_u}{1-\rho}$. The result then follows by using these bounds, noting that $\omega \sim \mathcal{N}(0, \text{blkdiag}(\Sigma))$, and applying Lemma **??** with the inequality $\|M\|_F \leq \sqrt{\text{rank}(M)}\|M\| \leq \sqrt{\max(n_1, n_2)}\|M\|$ for an $n_1 \times n_2$ matrix $M$. $\qquad \square$

We now proceed to prove our main regret upper bounds, for both the IIR and FIR case.

Let $\mathcal{E}_{\text{est},i}$ denote the event that the conclusions of Theorem **??** hold up to and including epoch $i$. Let $\{\widehat{\Phi}_{i,x}\}_{i \geq 0}$ and $\{\widehat{\Phi}_{i,u}\}_{i \geq 0}$ denote the closed loop SLS responses on the true system $(A_\star, B_\star)$. When

$\mathcal{E}_{\mathrm{est},i}$ holds, Theorem **??** in the IIR case and Theorem **??** in the FIR case state that uniformly for all epochs $i$ we have

$$\widehat{\mathbf{\Phi}}_{i,x} \in \mathcal{RH}_\infty(\widehat{C}, \widehat{\rho})\,, \;\; \widehat{\mathbf{\Phi}}_{i,u} \in \mathcal{RH}_\infty(\|K_\star\|\widehat{C}, \widehat{\rho})\,,$$

for

$$\widehat{C} = \frac{\mathcal{O}(1)C_\star}{(1-\rho_\star)^2}\,,$$
$$\widehat{\rho} = 7/8 + (1/8)\rho_\star\,,$$

in the IIR case and

$$\widehat{C} = \frac{\mathcal{O}(1)C_\star}{(1-\rho_\star)^3}\,,$$
$$\widehat{\rho} = 0.999 + 0.001\rho_\star\,,$$

in the FIR case. For ease of notation, define $\widehat{C}_c^2 := \frac{(\|Q\|+\|R\|\|K_\star\|)\widehat{C}^2}{(1-\widehat{\rho})^2}$.

Now fix an epoch $i \geq 1$ (the epoch $i = 0$ will be dealt with separately) and let $\mathbf{K}_i$ denote the controller that is active during epoch $i$. We invoke Lemma **??** conditioned on $\mathcal{E}_{\mathrm{est},i}$ and $x_{i,0}$ with $\delta \leftarrow \mathcal{O}(1)\delta/(i+1)^2$, $\Sigma \leftarrow \sigma_w^2 I + \sigma_{\eta,i}^2 B_\star B_\star^\top$, $C_x \leftarrow \widehat{C}$, $C_u \leftarrow \|K_\star\|\widehat{C}$, and $\rho \leftarrow \widehat{\rho}$. The conclusion is that with (conditional) probability at least $1 - \mathcal{O}(1)\delta/(i+1)^2$,

$$\sum_{k=1}^{T_i} x_{i,k}^\top Q x_{i,k} + u_{i,k}^\top R u_{i,k}$$

$$\leq J_T(A_\star, B_\star, \mathbf{K}_i; \sigma_w^2 I + \sigma_{\eta,i}^2 B_\star B_\star^\top)$$
$$+ \widehat{C}_c^2 \mathcal{O}\left(\|x_{i,0}\|_2^2 + (\sigma_w^2 + \sigma_{\eta,i}^2\|B_\star\|^2)(\sqrt{nT_i \log((i+1)/\delta)} + \log((i+1)/\delta))\right)$$

$$\leq T_i \left(1 + \frac{\sigma_{\eta,i}^2\|B_\star\|^2}{\sigma_w^2}\right) J(A_\star, B_\star, \mathbf{K}_i; \sigma_w^2 I)$$
$$+ \widehat{C}_c^2 \mathcal{O}\bigg(\sigma_w^2(n + \log((i+1)/\delta))\frac{\widehat{C}^2\widehat{\rho}^2(1 + \|B_\star\|)^2}{(1-\widehat{\rho})^2}$$
$$+ (\sigma_w^2 + \sigma_{\eta,i}^2\|B_\star\|^2)(\sqrt{nT_i \log((i+1)/\delta)} + \log((i+1)/\delta))\bigg)\,.$$

For the second inequality, we used the bound **(??)** and the bound on $\|x_{i,0}\|_2$ from **(??)**.

Furthermore, **(??)** and Theorem **??** in the IIR case (Theorem **??** in the FIR case) tell us that on $\mathcal{E}_{\mathrm{est},i}$, we have the sub-optimality bound

$$J(A_\star, B_\star, \mathbf{K}_i; \sigma_w^2 I) \leq (1 + C_{J_{i-1}})^2(1 + \mathcal{O}(1)\varepsilon_{i-1}(1 + \|K_\star\|)\|\mathfrak{R}_{A_\star + B_\star K_\star}\|_{\mathcal{H}_\infty})^2 J_\star\,,$$
$$\varepsilon_i = \widetilde{\mathcal{O}}\left(\frac{\sigma_w\|K_\star\|\widehat{C}}{\sigma_{\eta,i}}\sqrt{\frac{n+p}{T_i}}\right)\,.$$

Above, in the IIR case, we set $C_{J_i} = 0$ for all $i$, and in the FIR case we choose $C_{J_i} = 1/2^{i+1}$. Since $C_{J_i} \leq 1$, we have that $(1 + C_{J_i})^2 \leq 1 + 3C_{j_i}$. Recalling that $\sigma_{\eta,i}/\sigma_w = 2^{-i/6}$ and that $T_i = C_T 2^i$, we simplify $\varepsilon_i = \widetilde{\mathcal{O}}\left(\|K_\star\|\widehat{C}\sqrt{\frac{n+p}{C_T}}2^{-i/3}\right) := \widetilde{\mathcal{O}}(\frac{D_1}{\sqrt{C_T}}2^{-i/3})$ which gives us

$$(1 + \mathcal{O}(1)\varepsilon_{i-1}(1 + \|K_\star\|)\|\mathfrak{R}_{A_\star + B_\star K_\star}\|_{\mathcal{H}_\infty})^2$$
$$= 1 + \widetilde{\mathcal{O}}\left(\frac{D_1}{\sqrt{C_T}}(1 + \|K_\star\|)\|\mathfrak{R}_{A_\star + B_\star K_\star}\|_{\mathcal{H}_\infty}2^{-i/3} + \frac{D_1^2}{C_T}(1 + \|K_\star\|)^2\|\mathfrak{R}_{A_\star + B_\star K_\star}\|_{\mathcal{H}_\infty}^2 2^{-2i/3}\right)$$
$$:= 1 + \widetilde{\mathcal{O}}\left(\frac{D_2}{\sqrt{C_T}}2^{-i/3} + \frac{D_2^2}{C_T}2^{-2i/3}\right)\,.$$

This means that

$$T_i \left(1 + \frac{\sigma_{\eta,i}^2 \|B_\star\|^2}{\sigma_w^2}\right) J(A_\star, B_\star, \mathbf{K}_i; \sigma_w^2 I)$$

$$\leq T_i \left(1 + 2^{-i/3}\|B_\star\|^2\right)(1 + 3C_{J_{i-1}})\left(1 + \widetilde{\mathcal{O}}\left(\frac{D_2}{\sqrt{C_T}}2^{-i/3} + \frac{D_2^2}{C_T}2^{-2i/3}\right)\right)J_\star$$

$$\leq T_i \left(1 + \widetilde{\mathcal{O}}\left(\left(\frac{D_2}{\sqrt{C_T}} + \|B_\star\|^2\right)2^{-i/3} + \left(\frac{D_2^2}{C_T} + \frac{D_2\|B_\star\|^2}{\sqrt{C_T}}\right)2^{-2i/3} + \frac{D_2^2\|B_\star\|^2}{C_T}2^{-i}\right)\right)J_\star$$

$$+ \widetilde{\mathcal{O}}((1 + \|B_\star\|^2)(C_T + D_2\sqrt{C_T} + D_2^2)J_\star)$$

$$= T_i J_\star + \widetilde{\mathcal{O}}(\sqrt{C_T}D_2 + C_T\|B_\star\|^2)J_\star 2^{2i/3} + \widetilde{\mathcal{O}}(D_2^2 + \sqrt{C_T}D_2\|B_\star\|^2)J_\star 2^{i/3}$$

$$+ \widetilde{\mathcal{O}}((1 + \|B_\star\|^2)(C_T + D_2\sqrt{C_T} + D_2^2)J_\star).$$

Hence,

$$\sum_{k=1}^{T_i}(x_{i,k}^\top Q x_{i,k} + u_{i,k}^\top R u_{i,k} - J_\star)$$

$$\leq \widetilde{\mathcal{O}}(\sqrt{C_T}D_2 + C_T\|B_\star\|^2)J_\star 2^{2i/3} + \widetilde{\mathcal{O}}(D_2^2 + \sqrt{C_T}D_2\|B_\star\|^2)J_\star 2^{i/3}$$

$$+ \widetilde{\mathcal{O}}\left(\frac{\widehat{C}_c^2\sigma_w^2 n\widehat{C}^2(1 + \|B_\star\|)^2}{(1 - \widehat{\rho})^2}\right) + \widetilde{\mathcal{O}}(\widehat{C}_c^2\sigma_w^2\sqrt{nC_T}2^{i/2}) + \widetilde{\mathcal{O}}(\widehat{C}_c^2\sigma_w^2\|B_\star\|^2\sqrt{nC_T}2^{i/6})$$

$$+ \mathcal{O}(C_T 2^{i/2}(1 + \|B_\star\|^2)) + \widetilde{\mathcal{O}}((1 + \|B_\star\|^2)(C_T + D_2\sqrt{C_T} + D_2^2)J_\star).$$

On the other hand, when epoch $i = 0$, we have that

$$\sum_{k=1}^{T}x_{0,k}^\top Q x_{0,k} + u_{0,k}^\top R u_{0,k} \leq J_T(A_\star, B_\star, \mathbf{K}_0, \sigma_w^2 I + \sigma_{\eta,0}^2 B_\star B_\star^\top) + \widetilde{\mathcal{O}}(\widehat{C}_c^2\sigma_w^2(1 + \|B_\star\|^2)\sqrt{nC_T})$$

$$\leq C_T(1 + \|B_\star\|^2)J(A_\star, B_\star, \mathbf{K}_0, \sigma_w^2 I) + \widetilde{\mathcal{O}}(\widehat{C}_c^2\sigma_w^2(1 + \|B_\star\|^2)\sqrt{nC_T}).$$

Summing over all the epochs,

$$\mathsf{Regret}(T) = \sum_{i=0}^{O(\log_2 T)}\sum_{k=1}^{T_i}(x_{i,k}^\top Q x_{i,k} + u_{i,k}^\top R u_{i,k} - J_\star)$$

$$\leq \widetilde{\mathcal{O}}((\sqrt{C_T}D_2 + C_T\|B_\star\|^2)J_\star T^{2/3}) + \widetilde{\mathcal{O}}(\widehat{C}_c^2\sigma_w^2\sqrt{nC_T}T^{1/2})$$

$$+ \widetilde{\mathcal{O}}(D_2^2 + \sqrt{C_T}D_2\|B_\star\|^2 J_\star T^{1/3}) + \widetilde{\mathcal{O}}(\widehat{C}_c^2\sigma_w^2\|B_\star\|^2\sqrt{nC_T}T^{1/6})$$

$$+ \widetilde{\mathcal{O}}\left(\frac{\widehat{C}_c^2\sigma_w^2 n\widehat{C}^2(1 + \|B_\star\|)^2}{(1 - \widehat{\rho})^2} + C_T(1 + \|B_\star\|^2)J(A_\star, B_\star, \mathbf{K}_0, \sigma_w^2 I)\right)$$

$$+ \widetilde{\mathcal{O}}((1 + \|B_\star\|^2)(C_T + D_2\sqrt{C_T} + D_2^2)J_\star)$$

$$+ \widetilde{\mathcal{O}}(\widehat{C}_c^2\sigma_w^2(1 + \|B_\star\|^2)\sqrt{nC_T}) + \mathcal{O}(C_T(1 + \|B_\star\|^2)\sqrt{T}).$$

Using the bound on $C_T$ from (**??**), recalling that

$$D_2 = \sqrt{n + p}\|K_\star\|\widehat{C}(1 + \|K_\star\|)\|\mathfrak{R}_{A_\star + B_\star K_\star}\|_{\mathcal{H}_\infty},$$

and ignoring the $o(T^{2/3})$ terms in the regret bound, we have that the regret is bounded by in the IIR case

$$\widetilde{\mathcal{O}}\left((n + p)\|\mathfrak{R}_{A_\star + B_\star K_\star}\|_{\mathcal{H}_\infty}\frac{C_\star^3(1 + \|K_\star\|)^4}{(1 - \rho_\star)^6}J_\star T^{2/3} + (n + p)\frac{C_\star^4(1 + \|K_\star\|)^4\|B_\star\|^2}{(1 - \rho_\star)^8}J_\star T^{2/3}\right).$$

By using Lemma **??**, we have that $\|\mathfrak{R}_{A_\star + B_\star K_\star}\|_{\mathcal{H}_\infty} \leq \frac{C_\star}{1 - \rho_\star}$, and hence the bound in the IIR case simplifies to

$$\widetilde{\mathcal{O}}\left((n + p)\frac{C_\star^4(1 + \|K_\star\|)^4(1 + \|B_\star\|)^2 J_\star}{(1 - \rho_\star)^8}T^{2/3}\right).$$

Now for the FIR case, we use the bound (**??**) and ignoring the $o(T^{2/3})$ terms, the regret is bounded by

$$\widetilde{\mathcal{O}}\left( (n+p)\|\mathfrak{R}_{A_\star+B_\star K_\star}\|_{\mathcal{H}_\infty} \frac{C_\star^3(1+\|K_\star\|)^4}{(1-\rho_\star)^{11}} J_\star T^{2/3} + (n+p)\frac{C_\star^4(1+\|K_\star\|)^4\|B_\star\|^2}{(1-\rho_\star)^{16}} J_\star T^{2/3} \right) .$$

Using the same bound on $\|\mathfrak{R}_{A_\star+B_\star K_\star}\|_{\mathcal{H}_\infty}$ as before, we obtain the FIR regret bound

$$\widetilde{\mathcal{O}}\left( (n+p)\frac{C_\star^4(1+\|K_\star\|)^4(1+\|B_\star\|)^2}{(1-\rho_\star)^{16}} J_\star T^{2/3} \right) .$$

# E  Lower bound

This section is dedicated to proving Theorem 3.4. Throughout this section we assume the following setup and notation. We consider the LQR problem defined by

$$\min_{u_0,u_1,\ldots,u_{T-1}} \mathbb{E}\left[ x_T^\top P x_T + \sum_{t=0}^{T-1} u_t^\top R u_t + x_t^\top Q x_t \right] ,$$

$$\text{s.t. } x_{t+1} = A_\star x_t + B_\star u_t + w_t.$$

where $u_t$ is allowed to be any random variable taking values in $\mathbb{R}^p$ that is independent of the sigma algebra $\sigma(w_t, w_{t+1}, \ldots)$. In particular, $u_t$ can be a measurable function of $x_0, w_0, w_1, \ldots, w_{t-1}$, and possibly other exogenous randomness.

We assume that $Q$ and $R$ are both positive definite matrices. Throughout this section we denote by $P$ the solution to the discrete algebraic Riccati equation:

$$P_\star = A^\top P_\star A - A^\top P_\star B(R + B^\top P_\star B)^{-1}B^\top P_\star A + Q.$$

Moreover, we denote by $K_\star$ the optimal controller for the infinite horizon LQR problem, namely $K_\star = -(R + B^\top P_\star B)^{-1}B^\top P_\star A$. Hence, the optimal closed loop matrix is given by $M = A_\star + B_\star K_\star$. Throughout this section we assume that the system $(A, B)$ is controllable and hence $\rho(M) < 1$. Therefore, there exist $C > 0$ and $\rho \in (0,1)$ such that $\|M^k\|_2 \le C\rho^k$ for all $k \ge 1$.

The initial state $x_0$ for the LQR problem defined above is assumed to have distribution $\mathcal{N}(0, P_\infty)$, where $P_\infty$ is the unique solution to the Lyapunov equation

$$P_\infty = (A_\star + B_\star K_\star)P_\infty(A_\star + B_\star K_\star)^\top + \sigma_w^2 I_n.$$

The distribution $\mathcal{N}(0, P_\infty)$ corresponds to the stationary distribution of the optimal closed loop system $x_{t+1} = (A_\star + B_\star K_\star)x_t + w_t$. In particular, if $x_t \sim \mathcal{N}(0, P_\infty)$, then $x_{t+1} \sim \mathcal{N}(0, P_\infty)$.

We consider the objective

$$J_T(\nu_0, \nu_1, \ldots, \nu_{T-1}) = \mathbb{E}\left[ x_T^\top P_\star x_T + \sum_{t=0}^{T-1} u_t^\top R u_t + x_t^\top Q x_t \right], \qquad (\text{E.1})$$

where $u_t = K_\star x_t + \nu_t$ for the optimal controller $K_\star$. Then, since the terminal cost is given by $P_\star$, we know that the minimum of objective (**??**) over $\nu_0, \nu_1, \ldots, \nu_{T-1}$ such that $\nu_t$ is independent of $\sigma(w_t, w_{t+1}, \ldots)$ is achieved when all $\nu_t$ are identically zero. The random variables $\nu_t$ should be thought of as deviations from the optimal inputs $K_\star x_t$ for the infinite horizon LQR. Finally, since $x_0 \sim \mathcal{N}(0, P_\infty)$ we have that the optimal objective value is $J_T^\star = J_T(0) = TJ_\star + \mathbf{Tr}(P_\star P_\infty)$, where $J_\star = \sigma_w^2\,\mathbf{Tr}(P_\star)$ is the optimal objective value of the infinite horizon LQR.

The proof of Theorem 3.4 follows an argument inspired from the field of strongly convex optimization. We show that under the minimum eigenvalue condition of the process $z_t = [x_t^\top, u_t^\top]^\top$, the process $\{\nu_t\}_{t\ge 0}$ is bounded away from zero. Moreover, we show that the expected regret at time $T$ is a strongly convex function of $\nu_0, \nu_1, \ldots, \nu_{T-1}$, leading us to the desired conclusion. We proceed by proving a sequence of technical result, followed by the proof of Theorem 3.4.

**Lemma E.1.** . *Suppose that*

$$\lambda_{\min}\left( \sum_{t=0}^{T-1} \begin{bmatrix} x_t \\ u_t \end{bmatrix} \begin{bmatrix} x_t^\top & u_t^\top \end{bmatrix} \right) \ge \tau, \qquad (\text{E.2})$$

*with $u_t = K_\star x_t + \nu_t$. Then*

$$\sum_{t=0}^{T-1} \|\nu_t\|_2^2 \geq \left(1 + \sigma_{\min}(K_\star)^2\right) \tau \tag{E.3}$$

*Proof.* Consider $v = [v_1^\top, v_2^\top]^\top \in \mathbb{R}^{n+p}$ such that $\|v\|_2 = 1$ and $v_1 + K^\top v_2 = 0$ (such $v$ exists because $[I, K^\top]$ is an $n \times (n+p)$ matrix and hence has a non-trivial null space). Moreover, $\|v_2\|_2^2 \leq (1 + \sigma_{\min}(K_\star)^2)^{-1}$. Then, by assumption we have

$$\tau \leq \sum_{t=0}^{T-1} (\langle x_t, v_1 \rangle + \langle u_t, v_2 \rangle)^2 = \sum_{t=0}^{T-1} (\langle x_t, v_1 \rangle + \langle Kx_t + \nu_t, v_2 \rangle)^2 = \sum_{t=0}^{T-1} \langle \nu_t, v_2 \rangle^2$$

$$\leq \|v_2\|_2^2 \sum_{t=0}^{T-1} \|\nu_t\|_2^2 \leq \frac{1}{1 + \sigma_{\min}(K)^2} \sum_{t=0}^{T-1} \|\nu_t\|_2^2.$$

$\square$

**Lemma E.2.** *Denote by $M$ the optimal closed loop matrix $A_\star + B_\star K_\star$. Then*

$$J_T(\nu_0, \nu_1, \ldots, \nu_{T-1}) - J_T^\star = \mathbb{E}\left[\sum_{j=0}^{T-1} \nu_j^\top (B_\star^\top P_\star B_\star + R)\nu_j\right]$$

$$+ 2\mathbb{E}\left[\sum_{0 \leq i < j \leq T-1} \nu_i^\top B_\star^\top (M^\top)^{j-i} P_\star B_\star \nu_j\right].$$

*Proof.* We know that

$$J_T^* = \mathbb{E}\left[\sum_{t=0}^{T-1} x_{\star,t}^\top (Q + K_\star^\top RK_\star) x_{\star,t}\right] + \mathbb{E}\left[x_{\star,T}^\top P_\star x_{\star,T}\right],$$

where $x_{\star,t} = \sum_{j=-1}^{t-1} M^{t-1-j} w_j$. Here, $w_{-1} = x_0$ for convenience, and $w_t \overset{\text{i.i.d.}}{\sim} \mathcal{N}(0, \sigma_w^2 I_n)$ for convenience. Also,

$$J_T(\nu_0, \nu_1, \ldots, \nu_{T-1}) = \mathbb{E}\left[\sum_{t=0}^{T-1} x_t^\top Q x_t + (K_\star x_t + \nu_t)^\top R(K_\star x_t + \nu_t)\right] + \mathbb{E}\left[x_T^\top P_\star x_T\right],$$

where $x_t = \sum_{j=-1}^{t-1} M^{t-1-j} w_j + M^{t-1-j} B\nu_j$ and $\nu_{-1} = 0$. Recall that $\nu_t$ is independent of any $w_i$ with $i \geq t$. Hence, for any matrix $N$ we have that $\mathbb{E}\left[w_i^\top N \nu_t\right] = 0$ if $i \geq t$. Therefore

$$J_T - J_T^\star = \mathbb{E}\left[\sum_{t=0}^{T-1} \sum_{0 \leq i < j \leq t-1} 2w_i^\top (M^\top)^{t-1-i}(Q + K_\star^\top RK_\star)M^{t-1-j} B_\star \nu_j\right]$$

$$+ \mathbb{E}\left[\sum_{t=0}^{T-1} \sum_{i,j=0}^{t-1} \nu_i^\top B_\star^\top (M^\top)^{t-1-i}(Q + K_\star^\top RK_\star)M^{t-1-j} B_\star \nu_j\right] + \mathbb{E}\left[\sum_{t=0}^{T-1} \nu_t^\top R\nu_t\right]$$

$$+ \mathbb{E}\left[\sum_{t=0}^{T-1} \sum_{i=0}^{t-1} 2w_i^\top (M^\top)^{t-1-i} K_\star^\top R\nu_t\right]$$

$$+ \mathbb{E}\left[\sum_{0 \leq i < j \leq T-1} 2w_i^\top (M^\top)^{T-1-i} P_\star M^{T-1-j} B_\star \nu_j\right]$$

$$+ \mathbb{E}\left[\sum_{i,j=0}^{T-1} \nu_i^\top B_\star^\top (M^\top)^{T-1-i} P_\star M^{T-1-j} B_\star \nu_j\right].$$

Now, we note that the sum of the terms that depend linearly on $\nu_t$ is equal to zero, otherwise the optimum of $J_T$ would not be achieved at $\nu_t = 0$ for all $t$. Indeed, this can be checked through direct computation by remarking that the optimal controller $K_\star$ satisfies $K_\star^\top R = -M^\top P_\star B_\star$, and recalling that $P_\star$ satisfies the Lyapunov equation

$$P_\star = M^\top P_\star M + Q + K_\star^\top R K_\star. \tag{E.4}$$

Hence, we have

$$J_T - J_T^\star = \mathbb{E}\left[\sum_{j=0}^{T-2} \nu_j^\top B_\star^\top \left(\sum_{t=j+1}^{T-1} (M^\top)^{t-1-j}(Q + K^\top R K)M^{t-1-j}\right) B_\star \nu_j\right]$$

$$+ 2\mathbb{E}\left[\sum_{0 \leq i < j \leq T-2} \nu_i^\top B_\star^\top (M^\top)^{j-i}\left(\sum_{t=j+1}^{T-1} (M^\top)^{t-1-j}(Q + K_\star^\top R K_\star)M^{t-1-j}\right) B_\star \nu_j\right]$$

$$+ \mathbb{E}\left[\sum_{t=0}^{T-1} \nu_t^\top R \nu_t\right] + \mathbb{E}\left[\sum_{j=0}^{T-1} \nu_j^\top B_\star^\top (M^\top)^{T-1-j} P_\star M^{T-1-j} B_\star \nu_j\right]$$

$$+ 2\mathbb{E}\left[\sum_{0 \leq i < j \leq T-1} \nu_i^\top B_\star^\top (M^\top)^{T-1-i} P_\star M^{T-1-j} B_\star \nu_j\right].$$

The conclusion follows by using the Lyapunov equation (**??**) and simple algebra. □

**Lemma E.3.** *Let $M$ and $N$ be any matrices in $\mathbb{R}^{n \times n}$, with $N$ positive definite, and let $T$ be any positive integer. Also, consider the $(nT) \times (nT)$ block matrix $D(T)$ with blocks $D(T)_{i,j}$ equal to*

$$D_{i,j} = \begin{cases} (M^\top)^{j-i}\left(\sum_{k=0}^{T-j}(M^\top)^k N M^k\right) & \text{if } i < j, \\ \sum_{k=0}^{T-j}(M^\top)^k N M^k & \text{if } i = j, \\ \left(\sum_{k=0}^{T-i}(M^\top)^k N M^k\right) M^{i-j} & \text{if } i < j, \end{cases}$$

*where $1 \leq i, j \leq T$. The matrix $D$ is positive definite.*

*Proof.* We proceed by induction. Let $T = 2$. Then the matrix of interest is

$$D(2) = \begin{bmatrix} N + M^\top N M & M^\top N \\ NM & N \end{bmatrix}.$$

Since $N \succ 0$, we see that $D(T)$ is positive definite because its Schur complement is

$$N + M^\top N M - M^\top N N^{-1} N M = N \succ 0.$$

For $T > 2$ we proceed similarly. We consider the matrix $D(T)$ and take its Schur complement with respect to bottom right corner, i.e.

$$\begin{bmatrix} D(T)_{1,1} & \dots & D(T)_{1,T-1} \\ \vdots & \ddots & \vdots \\ D(T)_{T-1,1} & \dots & D(T)_{T-1,T-1} \end{bmatrix} - \begin{bmatrix} D(T)_{1,T} \\ \vdots \\ D(T)_{T-1,T} \end{bmatrix} D(T)_{T,T}^{-1} [D_{T,1} \quad \dots \quad D_{T,T-1}]$$

Let $i \leq j < T$. Then, the $(i, j)$ block of the Schur complement of $D(T)$ is

$$D(T)_{i,j} - D(T)_{i,T} D(T)^{-1} D(T)_{T,j} = (M^\top)^{j-i}\left(\sum_{k=0}^{T-j}(M^\top)^k N M^k\right) - (M^\top)^{T-i} N N^{-1} N M^{T-j}$$

$$= (M^\top)^{j-i}\left(\sum_{k=0}^{T-1-j}(M^\top)^k N M^k\right) = D(T-1)_{i,j}.$$

Similarly, if $j \leq i < T$ we have that $D(T)_{i,j} - D(T)_{i,T} D(T)^{-1} D(T)_{T,j} = D(T-1)_{i,j}$. Hence, we have shown that the Schur complement of $D(T)$ with respect to the entry $D(T)_{T,T}$ is $D(T-1)$. By induction this matrix is positive definite and the conclusion follows. □

**Lemma E.4.** *As before, $P_\star$ is the solution to the algebraic Riccati equation and $M = A_\star + B_\star K_\star$ is the optimal closed loop matrix. For any vectors $v_0$, $v_1$, ..., $v_{T-1}$ in $\mathbb{R}^p$ we have*

$$\sum_{j=0}^{T-1} v_j^\top (B_\star^\top P_\star B_\star + R) v_j + 2 \sum_{0 \le i < j \le T-1} v_i^\top B^\top (M^\top)^{j-i} P_\star B_\star v_j \ge \lambda_{\min}(R) \sum_{j=0}^{T-1} \|v_j\|_2^2.$$

*Proof.* It suffices to prove that the following matrix is positive semi-definite:

$$\begin{bmatrix} P_\star & M^\top P_\star & (M^\top)^2 P_\star & \cdots & (M^\top)^{T-1} P_\star \\ P_\star M & P_\star & M^\top P_\star & \cdots & (M^\top)^{T-2} P_\star \\ P_\star M^2 & P_\star M & P_\star & \cdots & (M^\top)^{T-2} P_\star \\ \vdots & \vdots & \vdots & \ddots & \vdots \\ P_\star M^{T-1} & P_\star M^{T-2} & P_\star M^{T-3} & \cdots & P_\star \end{bmatrix}.$$

The Schur complement of this matrix around the bottom right corner $P_\star$ has the form $D(T-1)$ with $N = Q + K_\star^\top R K_\star$, where $D(T-1)$ is defined as in Lemma **??**. To see this recall that $P_\star$ satisfies the Lyapunov equation (**??**). The conclusion follows. $\qquad\square$

**Lemma E.5.** *Fix a horizon $T_0 > 0$, and suppose the inputs are of the form $u_t = K_\star x_t + \nu_t$. Recall that there exists constants $C > 0$ and $\rho \in (0,1)$ such that $\|M^k\|_2 \le C\rho^k$ for all $k \ge 1$. Then*

$$\mathbb{E}\|x_{T_0}\|_2^2 \le 3C^2 \rho^{2T_0} \mathbb{E}\|x_0\|_2^2 + 3\frac{n\sigma_w^2 C^2}{1-\rho^2} + 3\frac{C^2}{1-\rho^2} \mathbb{E}\left[ \sum_{t=0}^{T_0} \|\nu_t\|_2^2 \right].$$

*Proof.* Recall that we denote by $M$ the closed loop matrix $A_\star + B_\star K_\star$. We have that

$$x_{T_0} = M^{T_0} x_0 + \sum_{t=0}^{T_0-1} M^{T_0-1-t}(B_\star \nu_t + w_t).$$

Then

$$\|x_{T_0}\|_2^2 \le 3\|M^{T_0} x_0\|_2^2 + 3\|\sum_{t=0}^{T_0-1} M^{T_0-1-t} w_t\|_2^2 + 3\|\sum_{t=0}^{T_0-1} M^{T_0-1-t} B\nu_t\|_2^2.$$

Recall that $\|M^t\|_2 \le C\rho^t$. Then

$$\mathbb{E}\|x_{T_0}\|_2^2 \le 3C^2 \rho^{2T_0} \mathbb{E}\|x_0\|_2^2 + 3\frac{n\sigma_w^2 C^2}{1-\rho^2} + 3\frac{C^2}{1-\rho^2} \mathbb{E}\left[ \sum_{t=0}^{T_0} \|\nu_t\|_2^2 \right].$$

$\qquad\square$

**Lemma E.6.** *Let $Q$ and $R$ be positive definite matrices, and $P_0 = 0$. Consider the Riccati recursion*

$$P_{t+1} = A^\top P_t A - A^\top P_t B (R + B^\top P_t B)^{-1} B^\top P_t A + Q.$$

*Then, if $P_\star$ is the unique solution of the Riccati equation, we have*

$$\|P_t - P_\star\|_2 \le \left(1 + \frac{1}{\nu}\right)^{-t}, \quad \text{where } \nu = 2\|P_\star\|_2 \max\left\{ \frac{\|A_\star\|_2^2}{\lambda_{\min}(Q)}, \frac{\|B_\star\|_2^2}{\lambda_{\min}(R)} \right\}.$$

*Moreover, we have that*

$$\sum_{t=0}^{\infty} \mathbf{Tr}(P_t) - \mathbf{Tr}(P_\star) \ge -n(1+\nu).$$

*Proof.* The first part follows from Proposition 1 of **?** ] on value iteration. The second part follows by bounding

$$\mathbf{Tr}(P_t) - \mathbf{Tr}(P_\star) \ge -n\|P_t - P_\star\|_2 \ge -n\left(1 + \frac{1}{\nu}\right)^{-t},$$

and summing up these inequalities. $\qquad\square$

**Lemma E.7.** *Fix a horizon $T_0 > 0$ and denote $\hat{x}_t = x_{t+T_0}$ and $\hat{u}_t = \hat{u}_{t+T_0}$. Then*

$$\mathbb{E}\left[\hat{x}_0^\top P \hat{x}_0\right] \leq \min_{\hat{u}_0, \hat{u}_1, \dots} \mathbb{E}\left[\sum_{t=0}^{T-1} \hat{x}_t^\top Q \hat{x}_t + \hat{u}_t^\top R \hat{u}_t\right] - TJ_\star + n\sigma_w^2(1+\nu) + \left(1 + \frac{1}{\nu}\right)^{-T} \mathbb{E}\|\hat{x}_0\|_2^2,$$

*where*

$$\nu = 2\|P_\star\|_2 \max\left\{\frac{\|A_\star\|_2^2}{\lambda_{\min}(Q)}, \frac{\|B_\star\|_2^2}{\lambda_{\min}(R)}\right\}.$$

*Proof.* Let us consider the Ricatti recursion

$$P_{t+1} = A^\top P_t A - A^\top P_t B(R + B^\top P_t B)^{-1} B^\top P_t A + Q,$$

where $P_0 = 0$. Then

$$\min_{\hat{u}_0, \hat{u}_1, \dots} \mathbb{E}\left[\sum_{t=0}^{T} \hat{x}_t^\top Q \hat{x}_t + \hat{u}_t^\top R \hat{u}_t\right] = \mathbb{E}\hat{x}_0^\top P_T \hat{x}_0 + \sigma_w^2 \sum_{t=0}^{T-1} \mathbf{Tr}\left(P_t\right).$$

From the first part of Lemma **??** we know that

$$\|P_T - P_\star\|_2 \leq \left(1 + \frac{1}{\nu}\right)^{-T},$$

while from the second part of that Lemma we know that

$$\sum_{t=0}^{T-1} \left[\mathbf{Tr}\left(P_t\right) - \mathbf{Tr}\left(P_\star\right)\right] \geq -n(1+\nu).$$

The conclusion follows once we recall that $J_\star = \sigma_w^2 \mathbf{Tr}(P_\star)$. $\qquad\square$

*Proof of Theorem 3.4.* Let $T_0 > 0$ to be chosen later and let $T \geq T_0$. We decompose the regret as the sum of the regret from 0 to $T - T_0 - 1$ and the regret from $T - T_0$ to $T - 1$, and we write the first component in terms terms of the expected cost $J_{T-T_0}$ defined in Eq. (**??**). We have

$$\sum_{t=0}^{T-1} \mathbb{E}\left[x_t^\top Q x_t + u_t^\top R u_t - J_\star\right] = \mathbb{E}\left[x_{T-T_0}^\top P_\star x_{T-T_0} + \sum_{t=0}^{T-T_0-1} x_t^\top Q x_t + u_t^\top R u_t\right] - J_{T-T_0}^\star$$

$$+ \sum_{t=T-T_0}^{T-1} \mathbb{E}\left[x_t^\top Q x_t + u_t^\top R u_t - J_\star\right] - T_0 J_\star$$

$$+ \mathbf{Tr}(P_\star P_\infty) - \mathbb{E}x_{T-T_0}^\top P_\star x_{T-T_0},$$

where we used $J_{T-T_0}^\star = (T - T_0)J_\star + \mathbf{Tr}(P_\star P_\infty)$. The term $\mathbf{Tr}(P_\star P_\infty)$ we an simply lower bound by zero since $P_\infty$ and $P_\star$ are positive semi-definite matrices. From Lemmas **??** and **??** we have

$$\mathbb{E}\left[x_{T-T_0}^\top P_\star x_{T-T_0} + \sum_{t=0}^{T-T_0-1} x_t^\top Q x_t + u_t^\top R u_t\right] - J_{T-T_0}^\star \geq \lambda_{\min}(R) \sum_{t=0}^{T-T_0-1} \|\nu_t\|_2^2.$$

By Lemma **??** we have that

$$\sum_{t=T-T_0}^{T-1} \mathbb{E}\left[x_t^\top Q x_t + u_t^\top R u_t - J_\star\right] - T_0 J_\star - \mathbb{E}x_{T-T_0}^\top P_\star x_{T-T_0}$$

$$\geq -n\sigma_w^2(1+\nu) - \left(1 + \frac{1}{\nu}\right)^{-T_0} \mathbb{E}\|x_{T-T_0}\|_2^2.$$

Then, from Lemma **??** we get

$$
\sum_{t=0}^{T-1} \mathbb{E}\left[x_t^\top Q x_t + u_t^\top R u_t - J_\star\right] \geq \frac{1}{2}\lambda_{\min}(R)\sum_{t=0}^{T-T_0}\|\nu_t\|_2^2
$$
$$
- \underbrace{\left(3C\rho^{2T_0}\,\mathbf{Tr}(P_\infty) + n\sigma_w^2\frac{\lambda_{\min}(R)}{2}\right)}_{C_0},
$$

by choosing

$$
T_0 \geq \frac{\log\left(\frac{2C^2}{(1-\rho^2)\lambda_{\min}(R)}\right)}{\log(1+\nu^{-1})}.
$$

The conclusion follows by Lemma **??**. $\qquad\square$

## F  Miscellaneous Results

First we state some results for the function class $\mathcal{RH}_\infty(C,\rho)$.

**Lemma F.1.** *Let $\mathbf{G}_i \in \mathcal{RH}_\infty(C_i,\rho_i)$ for $i = 1,2$ and Then $\mathbf{H} = \mathbf{G}_1\mathbf{G}_2 \in \mathcal{RH}_\infty(C,\rho)$ for any $\rho \in (\max(\rho_1,\rho_2),1)$ and $C = \max\left\{1, \frac{1}{e\log\left(\frac{\rho}{\max(\rho_1,\rho_2)}\right)}\frac{\rho}{\max(\rho_1,\rho_2)}\right\}C_1C_2$. Note for simplicity if we assume $\rho \geq 1/4$ we can take $C = \frac{6C_1C_2}{1-\rho}$ and $\rho = \mathsf{Avg}(\max(\rho_1,\rho_2),1)$.*

*Proof.* Assume wlog that $\rho_1 \geq \rho_2$. Note that $H(k) = \sum_{t=0}^k G_1(t)G_2(k-t)$, and therefore for all $k \geq 0$ we have that

$$
\|H(k)\| = \left\|\sum_{t=0}^k G_1(t)G_2(k-t)\right\| \leq C_1C_2\sum_{t=0}^k \rho_1^t\rho_2^{k-t}
$$
$$
\leq C_1C_2\sum_{t=0}^k \rho_1^k = C_1C_2(k+1)\rho_1^k,
$$

Fix a $\rho \in (\rho_1,1)$. Define $g(k) = (k+1)(\rho_1/\rho)^k$ and $h(k) = \log g(k)$. We see that $h'(k) = 0$ only for $k = k_* = \frac{1}{\log(\rho/\rho_1)} - 1$. Furthermore, $h(k_*) = \log(1/\log(\rho/\rho_1)) - 1 + \log(\rho/\rho_1)$. Hence, $g(k_*) = \frac{1}{e\log(\rho/\rho_1)}(\rho/\rho_1)$.

The claim now follows since for any $k \geq 0$,

$$
(k+1)\rho_1^k = (k+1)(\rho_1/\rho)^k\rho^k \leq \left[\sup_{k=0,1,\ldots}(k+1)(\rho_1/\rho)^k\right]\rho^k \leq \max\{1, g(k_*)\}\rho^k.
$$

We also use the inequality $\log(1+x) \geq x/2$ for $x \in [0,2.5]$. $\qquad\square$

**Lemma F.2.** *Let $\mathbf{G}_i \in \mathcal{RH}_\infty(C_i,\rho_i)$ for $i = 1,2$. Then $\mathbf{G}_1+\mathbf{G}_2 \in \mathcal{RH}_\infty(C_1+C_2, \max\{\rho_1,\rho_2\})$.*

*Proof.* Straightforward from triangle inequality and the definitions. $\qquad\square$

**Lemma F.3.** *Suppose that $\boldsymbol{\Delta} \in \mathcal{RH}_\infty(C,\rho)$ with $C \leq 2$ and $\rho \geq 1/e$, and furthermore $\|\boldsymbol{\Delta}\|_{\mathcal{H}_\infty} < 1$. Then we have*

$$
(I \pm \boldsymbol{\Delta})^{-1} \in \mathcal{RH}_\infty\left(1 + \frac{\mathcal{O}(1)C}{1-\rho}, \mathsf{Avg}(\rho,1)\right).
$$

*Proof.* The function $f(x) = \frac{x}{e \log(x)}$ is monotonically decreasing on the interval $(1, 1/\rho)$. Hence for any $x \in (1, 1/\rho)$, we have $f(x) \geq f(1/\rho) \geq f(e) = 1$. Applying the composition lemma (Lemma **??**) to the system $\mathbf{\Delta} \circ \mathbf{\Delta}$, we have that for $c_1 \in (1, 1/\rho)$,

$$\mathbf{\Delta}^2 \in \mathcal{RH}_\infty \left( \frac{c_1}{e \log(c_1)} C^2, c_1 \rho \right) .$$

Now if we recursively set $c_k \in (c_{k-1}, 1/\rho)$ for $k = 2, 3, ...$, repeated applications of the composition lemma yield that

$$\mathbf{\Delta}^n \in \mathcal{RH}_\infty \left( C^n \prod_{i=1}^{n-1} \frac{c_i}{e \log(c_i)}, c_{n-1}\rho \right) .$$

Let $c_\infty = \lim_{k \to \infty} c_k$, which exists and is finite because the sequence $c_k$ is monotonically increasing and bounded above. Furthermore, we have that for any $n \geq 2$,

$$C^n \prod_{i=1}^{n-1} \frac{c_i}{e \log(c_i)} \leq \left( \frac{Cc_\infty}{e} \right)^{n-1} \frac{C}{\log(\sum_{i=1}^{n-1} c_i)} \leq \left( \frac{Cc_\infty}{e} \right)^{n-1} \frac{C}{\log(c_1)} .$$

Now choose any strictly increasing sequence such that $c_\infty = \mathsf{Avg}(1, 1/\rho) = (1/2)(1/\rho + 1)$ and $c_1 = \mathsf{Avg}(1, c_\infty) = (1/4)(3 + 1/\rho)$. By the addition lemma (Lemma **??**), the assumption on $C$, and a simple limiting argument,

$$\sum_{n=0}^{\infty} \mathbf{\Delta}^n \in \mathcal{RH}_\infty (C', c_\infty \rho) ,$$

where $C'$ is given as

$$C' \leq 1 + C + \frac{C}{\log(c_1)} \frac{1}{1 - Cc_\infty/e} \leq 1 + C + \frac{2C}{\log(c_1)} .$$

The claim now follows by using the inequality $\log(1 + x) \geq x/2$ for $x \in [0, 2.5]$ and the assumed bound $C \leq 2$. $\qquad\square$

**Lemma F.4.** *Suppose that* $\mathbf{G} \in \mathcal{RH}_\infty(C, \rho)$. *Then* $\|\mathbf{G}\|_{\mathcal{H}_\infty} \leq \frac{C}{1-\rho}$.

*Proof.* We have that

$$\|\mathbf{G}\|_{\mathcal{H}_\infty} = \sup_{z \in \mathbb{T}} \|\mathbf{G}(z)\| = \sup_{z \in \mathbb{T}} \left\| \sum_{k=0}^{\infty} G(k) z^{-k} \right\| \leq C \sum_{k=0}^{\infty} \rho^k = \frac{C}{1 - \rho}.$$

$\qquad\square$

Next, a probabilistic lemma which we use to control the LQR cost on a finite horizon.

**Lemma F.5.** *Let $x$ and $M$ be fixed, and $w \sim \mathcal{N}(0, \Sigma)$, with $\Sigma \succ 0$ and $\|\Sigma\| = \sigma^2$. Then there exists a universal constant $c > 0$ such that with probability at least $1 - \delta$*

$$\begin{bmatrix} x \\ w \end{bmatrix}^\top M \begin{bmatrix} x \\ w \end{bmatrix} \leq x^\top M_{11} x + 2\sqrt{2}\sigma \|x\| \|M_{12}\| \sqrt{\log\left(\frac{2}{\delta}\right)}$$

$$+ \mathbf{Tr}\, M_{22}\Sigma + c\sigma^2 \|M_{22}\|_F \sqrt{\log\left(\frac{2}{\delta}\right)} + c\sigma^2 \|M_{22}\| \log\left(\frac{2}{\delta}\right). \qquad (\text{F.1})$$

*Proof.* Expanding the quadratic we have

$$\begin{bmatrix} x \\ w \end{bmatrix}^\top M \begin{bmatrix} x \\ w \end{bmatrix} = x^\top M_{11} x + 2x^\top M_{12} w + w^\top M_{22} w.$$

Noting that $x^\top M_{12} w \sim \mathcal{N}(0, x^\top M_{12} \Sigma M_{12}^\top x)$, by standard Gaussian concentration we have with probability at least $1 - \frac{\delta}{2}$ that

$$
\begin{aligned}
x^\top M_{12} w &\leq \sqrt{2 x^\top M_{12} \Sigma M_{12}^\top x \log\left(\tfrac{2}{\delta}\right)} \\
&\leq \sqrt{2} \|x\| \|M_{12}\| \|\Sigma\|^{\frac{1}{2}} \sqrt{\log\left(\tfrac{2}{\delta}\right)} \\
&= \sqrt{2} \|x\| \sigma \|M_{12}\| \sqrt{\log\left(\tfrac{2}{\delta}\right)}.
\end{aligned}
$$

On the other hand, by the Hanson-Wright inequality [**?** ], we have that with probability at least $1 - \frac{\delta}{2}$ that

$$
\begin{aligned}
w^\top M_{22} w &\leq \mathbf{Tr}\, M_{22}\Sigma + c\sqrt{\|\Sigma^{\frac{1}{2}} M_{22} \Sigma^{\frac{1}{2}}\|_F^2 \log\left(\tfrac{2}{\delta}\right)} + c\|\Sigma^{\frac{1}{2}} M_{22} \Sigma^{\frac{1}{2}}\|^2 \log\left(\tfrac{2}{\delta}\right) \\
&\leq \mathbf{Tr}\, M_{22}\Sigma + c\sigma^2 \|M_{22}\|_F \sqrt{\log\left(\tfrac{2}{\delta}\right)} + c\sigma^2 \|M_{22}\| \log\left(\tfrac{2}{\delta}\right).
\end{aligned}
$$

$\square$

**Lemma F.6.** *Let $\Sigma$ be a $n \times n$ positive-definite matrix and let $K$ be a real $p \times n$ matrix. Then, for any $\sigma_u \in \mathbb{R}$ we have that*

$$
\lambda_{\min}\left(\begin{bmatrix} \Sigma & \Sigma K^\top \\ K\Sigma & K\Sigma K^\top + \sigma_u^2 I \end{bmatrix}\right) \geq \sigma_u^2 \min\left(\frac{1}{2}, \frac{\lambda_{\min}(\Sigma)}{2\|K\Sigma K^\top\|_2 + \sigma_u^2}\right).
$$

*Proof.* We find $0 < \gamma_1 < 1$ and $\gamma_2 > 0$ such that the following condition holds

$$
\begin{bmatrix} \Sigma & \Sigma K^\top \\ K\Sigma & K\Sigma K^\top + \sigma_u^2 I \end{bmatrix} \succeq \begin{bmatrix} \gamma_1 \Sigma & 0 \\ 0 & \gamma_2 I \end{bmatrix}.
$$

By Schur complements, this condition is equivalent to

$$
\begin{aligned}
0 &\preceq K\Sigma K^\top + (\sigma_u^2 - \gamma_2)I - K\Sigma((1 - \gamma_1)\Sigma)^{-1}\Sigma K^\top \\
&= -\frac{\gamma_1}{1 - \gamma_1} K\Sigma K^\top + (\sigma_u^2 - \gamma_2)I.
\end{aligned}
$$

Now set $\gamma_2 = \sigma_u^2/2$ and $\gamma_1 = \frac{\sigma_u^2}{2\|K\Sigma K^\top\|_2 + \sigma_u^2}$. $\square$

# G  Implementation of Adaptive Methods

We consider several adaptive methods for numerical comparison. This section described the relevant implementation details.

## G.1  Optimism in the Face of Uncertainty

At the start of each epoch, the OFU method computes a confidence set around the dynamics and then finds the $(A, B)$ that would achieve the smallest LQR cost. The method then plays the associated optimal controller.

The confidence sets at epoch $i$ are of the form

$$
C_i(\varepsilon) = \{\Theta \in \mathbb{R}^{n \times (n+p)} : \mathbf{Tr}((\Theta - \widehat{\Theta}_i) Z_{T_i} (\Theta - \widehat{\Theta}_i)^\top) \leq \varepsilon\},
$$

$$
Z_{T_i} = \lambda I + \sum_{i=1}^{T_i} \begin{bmatrix} x_t \\ u_t \end{bmatrix} \begin{bmatrix} x_t \\ u_t \end{bmatrix}^\top. \tag{G.1}
$$

Here, $\widehat{\Theta}_i$ denotes the (regularized) least squares estimate of the true parameters $\Theta_* = (A_\star, B_\star)$. For our experiments, we set $\lambda = 10^{-5}$ and $\varepsilon = \mathbf{Tr}((\widehat{\Theta}_i - \Theta_*) Z_{T_i} (\widehat{\Theta}_i - \Theta_*)^\top)$ using the true and estimation values of $(A, B)$.

Then controller is selected by finding the "best" dynamics. To be precise, let $J(A, B) = \mathbf{Tr}(P(A, B))$, where $P(A, B)$ is the solution to the discrete algebraic Riccati solution

$$P = A^\top P A - A^\top P B (B^\top P B + R)^{-1} B^\top P A + Q .$$

Then for every epoch of OFU, it is necessary to solve to the non-convex optimization problem

$$[\widetilde{A}, \widetilde{B}] = \arg \min_{[A,B] \in C_i(\varepsilon)} J(A, B) . \tag{G.2}$$

up to an absolute error of at most $O(1/\sqrt{T_i})$.

As in Section 5.4 of [? ], we heuristically solve this optimization problem using projected gradient descent (PGD). An expression for the gradient of $\Theta \mapsto J(A, B)$ is derived in [? ] (see also [4]) by use of the implicit function theorem. Specifically, $\nabla_\Theta \mathbf{Tr}(P(A, B))$ evaluated at a point $\Theta = (A, B)$ is an $n \times (n + p)$ matrix $D$. The $i, j$-th entry is given by $\mathbf{Tr}(E_{ij})$, where $E_{ij}$ is the solution to the Lyapunov equation

$$E_{ij} = A_c^\top E_{ij} A_c + 2\mathrm{Sym}\left( A_c^\top P(A, B) e_i e_j^\top \begin{bmatrix} I \\ K \end{bmatrix} \right) ,$$

with $K$ as the optimal LQR controller for $(A, B)$, $A_c = A + BK$, and $\mathrm{Sym}(A) = \frac{1}{2}(A + A^\top)$. Finally, the projection of $\Theta$ onto the set $C_i(\varepsilon)$ can be solved by a eigendecomposition of $\check{Z}_{T_i}$ followed by a scalar root-finding search. The details of this are also found in Section 5.4 of [? ].

We determine the end of an epoch using a switching rule based on a slight modification of the determinant condition of [1]. We switch an epoch when both (a) $T - T_i \geq 10$ and (b) $\det(Z_T) > 2 \det(Z_{T_i})$ hold. The first condition is to ensure that the switches are not too frequent in the beginning of the algorithm.

## G.2 Thompson Sampling

The Thompson sampling algorithm is nearly identical to the OFU algorithm, except the optimization problem (??) is replaced by sampling. While the description of Thompson sampling in the Bayesian setting of [2] and [16] requires sampling from the posterior distribution, we follow the more frequentist setting of [4] and sample a point $\widetilde{\Theta}$ uniformly at random from the confidence set $C_i(\varepsilon)$ as in (??).

We implement this uniform samping by first drawing a $U \sim \mathrm{Unif}([0, 1])$ and a $\eta \in \mathbb{R}^{n \times (n+p)}$ with each $\eta_{ij} \sim \mathcal{N}(0, 1)$, and setting

$$\widetilde{\Theta} = \widehat{\Theta} + \sqrt{\varepsilon} \left( \frac{U^{1/(n(n+p))}}{\|\eta\|_F} \eta \right) Z_{T_i}^{-1/2} .$$

For the epoch switching rule, we follow the suggestion of [4] to force exploration after $\tau$ iterations, where we set $\tau = 500$. Specifically, we switch an epoch when the following predicate holds:

$$(T - T_i \geq \tau) \text{ or } ((T - T_i \geq 10) \text{ and } (\det(Z_T) > 2 \det(Z_{T_i}))) .$$

## G.3 Robust Adaptive Control with FIR truncation

We now describe how to turn the infinite-dimensional optimization problem in Algorithm 1 into a finite-dimensional problem. First, recall the problem we want to solve,

$$\mathrm{minimize}_{\gamma \in [0,1)} \frac{1}{1 - \gamma} \min_{\mathbf{\Phi}_x, \mathbf{\Phi}_u, V} \left\| \begin{bmatrix} Q^{1/2} & 0 \\ 0 & R^{1/2} \end{bmatrix} \begin{bmatrix} \mathbf{\Phi}_x \\ \mathbf{\Phi}_u \end{bmatrix} \right\|_{\mathcal{H}_2}$$

$$\text{s.t. } \begin{bmatrix} zI - \widehat{A} & -\widehat{B} \end{bmatrix} \begin{bmatrix} \mathbf{\Phi}_x \\ \mathbf{\Phi}_u \end{bmatrix} = I + \frac{1}{z^F} V , \quad \frac{\sqrt{2}\varepsilon}{1 - C_x \rho^{F+1}} \left\| \begin{bmatrix} \mathbf{\Phi_x} \\ \mathbf{\Phi_u} \end{bmatrix} \right\|_{\mathcal{H}_\infty} \leq \gamma , \quad \text{(G.3)}$$

$$\|V\| \leq C_x \rho^{F+1} , \quad \mathbf{\Phi}_x \in \frac{1}{z} \mathcal{RH}_\infty^F(C_x, \rho) , \quad \mathbf{\Phi}_u \in \frac{1}{z} \mathcal{RH}_\infty^F(C_u, \rho) .$$

Ignoring the outer minimization over $\gamma$ (which can be solved with bisection), the inner minimization is convex. Truncating the system responses to be FIR of length $F$ means that

$$\mathbf{\Phi}_x = \sum_{k=1}^{F} \Phi_x(k) z^{-k} , \quad \mathbf{\Phi}_u = \sum_{k=1}^{F} \Phi_u(k) z^{-k} .$$

All pieces of the infinite dimensional problem can be written in terms of these variables. First, consider the $\mathcal{H}_2$ cost in the objective. By Parseval's identity, we can simply add the second order cone constraint

$$\left\| \begin{bmatrix} Q^{1/2}\Phi_x(1) \\ \vdots \\ Q^{1/2}\Phi_x(F) \\ R^{1/2}\Phi_u(1) \\ \vdots \\ R^{1/2}\Phi_u(F) \end{bmatrix} \right\|_F \leq t\,, \tag{G.4}$$

and minimize $t$. Next, we consider the constraints of the original optimization. The function space constraints reduce to the requirement that

$$\|\Phi_x(k)\| \leq C_x \rho^k\,, \quad \|\Phi_u(k)\| \leq C_u \rho^k\,, \quad k = 1, ..., F\,. \tag{G.5}$$

Next, to rewrite the subspace constraint, we first consider that

$$z\mathbf{\Phi}_x = \sum_{k=0}^{F-1} \Phi_x(k+1) z^{-k}\,,$$

then the subspace constraint yields the following equality constraints,

$$\begin{aligned} \Phi_x(1) &= I\,, \\ \Phi_x(k+1) &= \widehat{A}\Phi_x(k) + \widehat{B}\Phi_u(k)\,, \quad k = 1, ..., F-1\,, \\ V &= \widehat{A}\Phi_x(F) + \widehat{B}\Phi_u(F)\,. \end{aligned} \tag{G.6}$$

The only constraint that remains is the $\mathcal{H}_\infty$ constraint, for which we use the following result.

**Theorem G.1** (Theorem 5.8, [**?** ]). *Consider the $T$-length FIR filter*

$$\mathbf{H}(z) = \sum_{k=0}^{T} H_k z^{-k}\,, H_k \in \mathbb{R}^{p \times m}\,.$$

*Define the matrix*

$$\overline{H} = \begin{bmatrix} H_0 \\ \vdots \\ H_T \end{bmatrix} \in \mathbb{R}^{p(T+1) \times m}\,.$$

*We have that $\|\mathbf{H}(z)\|_{\mathcal{H}_\infty} \leq \gamma$ iff there exists $Q = Q^\top \succeq 0$ with $Q \in \mathbb{R}^{p(T+1) \times p(T+1)}$ satisfying*

$$Q = \begin{bmatrix} Q_{00} & Q_{01} & \dots & Q_{0T} \\ * & Q_{11} & \dots & Q_{1T} \\ * & * & \ddots & \vdots \\ * & * & * & Q_{TT} \end{bmatrix}\,, \quad Q_{ij} \in \mathbb{R}^{p \times p}\,,$$

$$\sum_{t=0}^{T} Q_{tt} = \gamma^2 I_p\,, \quad \sum_{t=0}^{T-k} Q_{t(t+k)} = 0_{p \times p}\,, k = 1, ..., T\,, \quad \begin{bmatrix} Q & \overline{H} \\ \overline{H}^\top & I_m \end{bmatrix} \succeq 0\,.$$

For the SLS problem, the $\mathcal{H}_\infty$ constraint on is the filter

$$\mathbf{H}(z) = \sum_{k=1}^{F} \begin{bmatrix} \Phi_x(k) \\ \Phi_u(k) \end{bmatrix} z^{-k}\,.$$

The constraint can be rewritten using the LMI in Theorem **??**. To avoid a decision variable of size $(n+p)(F+1) \times (n+p)(F+1)$, we instead consider the transpose system $\mathbf{H}^\top$ which has the same $\mathcal{H}_\infty$ norm and coefficients of size $n \times (n+p)$.

**(a)** Cumulative Regret        **(b)** Infinite Horizon LQR Cost

**Figure 3:** A comparison of different adaptive methods on 500 experiments of the large-transient system example (**??**). In (a), the median and 90th percentile cumulative regret is plotted over time. In (b), the median and 90th percentile infinite-horizon LQR cost of the epoch's controller.

Putting this together, we arrive at the following SDP, which can be solved using an off the shelf solver,

$$
\min_{\substack{\mathbf{\Phi}_x[k]\in\mathbb{R}^{n\times n},\ \mathbf{\Phi}_u[k]\in\mathbb{R}^{p\times n},\ V\in\mathbb{R}^{n\times n}\\ P\in\mathbb{R}^{n(F+1)\times n(F+1)},\ t\in\mathbb{R}}} t
$$

$$
\text{s.t.} \quad (\textbf{??})\,,\ (\textbf{??})\,,\ (\textbf{??})\,,
$$

$$
\sum_{t=0}^{F} P_{tt} = \gamma^2 I\,,\quad \sum_{t=0}^{F-k} P_{t(t+k)} = 0\,,\quad k=1,...,F\,,
$$

$$
\overline{H} = \frac{\sqrt{2}\varepsilon}{1-C_x\rho^{F+1}}\begin{bmatrix}0_{n\times n} & 0_{n\times p}\\ \Phi_x(1)^\top & \Phi_u(1)^\top\\ \vdots & \vdots\\ \Phi_x(F)^\top & \Phi_u(F)^\top\end{bmatrix}\,,\quad \begin{bmatrix}P & \overline{H}\\ \overline{H}^\top & I_m\end{bmatrix}\succeq 0\,,
$$

$$
\|V\| \le C_x\rho^{F+1}\,.
$$

For our experiments, we used the SCS solver [**?** ] via CVXPY [**?** ].

Finally, once the FIR responses $\{\Phi_x(k)\}_{k=1}^{F}$ and $\{\Phi_u(k)\}_{k=1}^{F}$ are found, we need a way to implement the system responses as a controller. We represent the dynamic controller $\mathbf{K} = \mathbf{\Phi}_u\mathbf{\Phi}_x^{-1}$ by finding an equivalent state-space realization $(A_K, B_K, C_K, D_K)$ via Theorem 2 of [**?** ].

As a final note, the adaptive method as described in Algorithm 1 requires several constants to be specified. For the numerical experiments, we set $\sigma_{\eta,i} = C_\eta\sigma_w T_i^{-1/3}$ where we vary $C_\eta$ for different experiments, fix $\gamma = 0.98$, and use a fixed FIR trunction length of $F = 12$. For the experiments in Section 4, we set $C_\eta = 0.1$.

# H   Additional Experiments

## H.1   Large-Transient Dynamics

We present the regret comparison results using another system

$$
A_\star = \begin{bmatrix}2 & 0 & 0\\ 4 & 2 & 0\\ 0 & 4 & 2\end{bmatrix}\,,\quad B_\star = I,\ Q = 10I,\ R = I\,. \tag{H.1}
$$

The system is both unstable and has large transients. Each state receives direct input, and the cost is such that input size is penalized relatively less than state. This problem setting is amenable to robust methods due to both the cost ratio and the large transients, which are factors that may hurt optimistic methods. For this experiment, we ran all adaptive methods as described in Appendix **??**, and used an initialization with a horizon of length $T_0 = 250$ and $C_\eta = 2$.

The performance of the various adaptive methods is compared in Figure **??**. The median and 90th percentile regret over 500 instances is displayed in Figure **??**a, which gives an idea of both "average"

| (a) OFU | (b) TS | (c) Robust |

**Figure 4:** A comparison of cumulative regret when enlarged error bounds are used for synthesis, rather than the true errors. Both the median over 500 trials and the 90th percentile regret are plotted. In (a) is OFU, in (b) is TS, in (c) is robust. The plots show modest if any degradation in performance.

and worst-case behavior. Overall, the methods have very similar performance. One benefit of robustness is the guaranteed stability and therefore bounded infinite-horizon cost at every point during operation. In Figure **??**b, this infinite-horizon cost of the controller in each epoch is plotted. This measures the cost of using each epoch's controller indefinitely, rather than continuing to update its parameters. Especially for small numbers of iterations, the robust method performs relatively better than other adaptive algorithms, indicating that it is more amenable to early stopping.

## H.2 Error Scaling

In our experiments, we use the actual estimation errors for controller synthesis. To examine the effect of this choice, we artificially inflate the estimation errors by various multipliers, and plot the regret for various methods in Figure **??**. These experiments were run on the the graph Laplacian example in (4.1) with an initialization with a horizon of length $T_0 = 300$ and $C_\eta = 1$.

The adaptive methods were run as described in Appendix **??**. The error term $\varepsilon$ for OFU and TS appears in the computation of the uncertainty set as in (**??**). The errors $\varepsilon_A$ and $\varepsilon_B$ for the robust adaptive method appear in (**??**). The plot shows a modest degradation in regret as these terms are increased.

## H.3 Learning the Disturbance Process

We consider the problem of regulating a known system which is subject to disturbances correlated in time. These disturbances are modeled as the output of a LTI filter driven by white noise. In other words,

$$x_{k+1} = A_\star x_k + B_\star u_k + d_k , \qquad d_{k+1} = A_d d_k + w_k,$$

where $x_k$ is the state to drive to zero, and $d_k$ are the disturbances. We will take $(A_\star, B_\star)$ to be known and $A_d$ unknown. This setting models many phenomenon related to demand forecasting, in which the dynamics of e.g. a server farm is known, and the changes in demand are stochastic but correlated in time, and can thus be approximated by the output of an LTI filter.

The plant inputs $u_k$ are designed for regulation. The controller design problem can be formulated as an optimization problem by defining the augmented system as

$$\begin{bmatrix} x_{k+1} \\ d_{k+1} \end{bmatrix} = \begin{bmatrix} A_\star & I \\ 0 & A_d \end{bmatrix} \begin{bmatrix} x_k \\ d_k \end{bmatrix} + \begin{bmatrix} B_\star \\ 0 \end{bmatrix} u_k + \begin{bmatrix} 0 \\ I \end{bmatrix} w_k . \tag{H.2}$$

We will denote the augmented state $z_k = [x_k; d_k]$. Then the control actions can be designed using an adaptive LQR strategy. In many situations, inputs are relatively more costly, corresponding for example to energy usage. Defining an LQR cost directly related to the economics of the system can be unwise, due to the resulting tendency for states to become large, which may correspond to unsafe execution. While tuning the quadratic cost to represent a mixture of economic and safety considerations can often achieve good behavior in practice, the method is heuristic and lacks guarantees. Instead, consider the explicit addition of a constraint on the state, $\|x_k\|_\infty \leq a$ for $0 \leq k \leq H$ for some horizon (which may be infinite).

To state the necessary modification to the controller synthesis problem, we define the norm

$$\|\mathbf{M}\|_{\mathcal{L}_1} = \sup_{\|\mathbf{w}\|_\infty = 1} \|\mathbf{Mw}\|_\infty \,,$$

for both system responses and state matrices. This norm corresponds to the $\ell_\infty \mapsto \ell_\infty$ operator norm.

**Proposition H.1.** *For the system described in (**??**), let $\mathbf{\Phi}_z$ denote a closed-loop state response. Then consider constraints*

$$\|(\mathbf{\Phi}_z)_{22}\|_{\mathcal{L}_1} \leq \gamma/\tilde{\varepsilon}_A \,,$$
$$\|(\mathbf{\Phi}_z)_{12}\|_{\mathcal{L}_1} \leq \frac{a}{b} \cdot (1 - \gamma) := c \tag{H.3}$$

*where $(\mathbf{\Phi}_z)_{ij}$ denotes the blocks defined by the partition of $z_t$ into $x_t$ and $d_t$, and $\|\widehat{A}_d - A_d\|_{\mathcal{L}_1} \leq \tilde{\varepsilon}_A$. The addition of these constraints to the synthesis problem in (**??**) ensures that the resulting closed loop system has $\|x_k\|_\infty \leq a$ for $0 \leq k \leq H$ as long as $\|w_k\|_\infty \leq b$ for $0 \leq k \leq H$.*

*Proof.* In transfer function notation, the state of the plant can be described by

$$\mathbf{x} = \begin{bmatrix} I & 0 \end{bmatrix} \mathbf{z} = \begin{bmatrix} I & 0 \end{bmatrix} \mathbf{\Phi}_z (I + \widehat{\mathbf{\Delta}})^{-1} \begin{bmatrix} 0 \\ I \end{bmatrix} \mathbf{w} \,.$$

Furthermore, due to the known structure of the dynamics,

$$(I + \widehat{\mathbf{\Delta}})^{-1} = \left( I + \begin{bmatrix} 0 & 0 \\ 0 & \Delta_A \end{bmatrix} \mathbf{\Phi}_z \right)^{-1} = \begin{bmatrix} I & 0 \\ X & (I + \Delta_A(\mathbf{\Phi}_z)_{22})^{-1} \end{bmatrix} \,,$$

where $X = (I + \Delta_A(\mathbf{\Phi}_z)_{22})^{-1} \Delta_A(\mathbf{\Phi}_z)_{21}$. Then we have, letting $\widehat{\mathbf{\Delta}}_{22} = \Delta_A(\mathbf{\Phi}_z)_{22}$,

$$\mathbf{x} = \begin{bmatrix} I & 0 \end{bmatrix} \mathbf{\Phi}_z \begin{bmatrix} 0 \\ (I + \widehat{\mathbf{\Delta}}_{22})^{-1} \end{bmatrix} \mathbf{w} = (\mathbf{\Phi}_z)_{12}(I + \widehat{\mathbf{\Delta}}_{22})^{-1} \mathbf{w} \,.$$

Finally, to bound the size of the state,

$$\|\mathbf{x}\|_\infty \leq \|(\mathbf{\Phi}_z)_{12}(I + \widehat{\mathbf{\Delta}}_{22})^{-1}\|_{\mathcal{L}_1} \|\mathbf{w}\|_\infty \leq \frac{1}{1 - \|\widehat{\mathbf{\Delta}}_{22}\|_{\mathcal{L}_1}} \|(\mathbf{\Phi}_z)_{12}\|_{\mathcal{L}_1} \|\mathbf{w}\|_\infty \,.$$

Then we have that $\|\widehat{\mathbf{\Delta}}_{22}\|_{\mathcal{L}_1} \leq \tilde{\varepsilon}_A \|(\mathbf{\Phi}_z)_{22}\|_{\mathcal{L}_1}$, so the result follows from the constraints and the assumption on $w_k$. $\square$

Therefore, with either a bounded noise assumption on $w_k$ or a high-probability bound over a finite time horizon, we can apply the previous result to synthesize safe controllers. In the example displayed in Figure 2, the constraint as in (**??**) is added to the controller synthesis procedure with $c = 0.1$ and $\gamma = 0.98$.