[Reviews · NeurIPS 2018]

Reviewer 1



The paper studies LQR synthesis for LTI systems with unknown dynamics matrices. The central element of the paper is a (novel) algorithm that utilizes a convex optimization approach (the so-called System Level Synthesis approach, SLS) for synthesizing LQR controllers using estimated dynamics models. The SLS approach allows for an analysis of how the error in the matrix estimation affects the regret of the LQR controller. Using this controller synthesis, upper bounds on the estimation error of the dynamics matrices as well as upper and lower bounds for the expected loss are provided. The method is compared to existing approaches on a benchmark system. This computational study shows a comparable performance of all methods, with the presented method giving the nicest theoretical guarantees (e.g. stability). Quality: The paper is well and carefully written. It takes a very formal approach to the problem and presents an extensive mathematical analysis. Various mathematical concepts and approximation bounds are used to derive the insightful bounds. Unfortunately, I did not get a good intuition on the overall argument leading to the result and I would appreciate some more informal explanations on how the bounds are derived. Some aspects are confusing in the paper are confusing. For example, Algorithm 1 requires to specify failure probability $\delta$. However, this probability seems to be never used. Such details should be clarified. Clarity: The paper is carefully written. However, clarity is a major concern about the paper. I appreciate the authors' effort to move the math heavy analysis to the appendix. However, the main body cannot be read without the appendix. For example, the semidefinite optimization problem for the controller synthesis appears only in the pseudo-code of Algorithm 1. The reader needs to read the appendix, to understand what happens here. It would be nice to have a brief discussion of the optimization problem in Section 2. I would prefer such a discussion over the current - very shortened - sketch of System Level Synthesis. Although the general argumentation of the paper is clear, a outline of the overall proof is missing. The theoretical analysis is fairly heavy, using many approximations and bounds. The paper would become much better accessible for a reader, if the mathematical steps leading to the result were informally described initially. At some points in the long analysis, the notation is not carefully enough introduced. For example, in Theorem B1 $\frac{R}$ is used without being introduced. Also the notation $\frac{R}(1 : F)$ in proof of Lemma B.2 is not explained. As another example, in line 386 "e" is introduced but not used later on (might be a typo). Such details make it unnecessary hard for the reader to follow the paper. Overall, the paper is very long and dense. It might be beneficial for the readability of the paper to further limit the scope of the analysis and the discussion. Originality: The paper studies an interesting problem and provides nice theoretical results. I like use of robust control synthesis for building a robust adaptive control scheme. Without claiming to know all relevant literature, I found the derived bounds insightful and interesting. Significance: The paper makes an interesting theoretical contribution by providing an algorithm for the adaptive LQR problem with transparent theoretical guarantees. It is hard for me to judge the practical relevance of the method. Although the proposed method is polynomial time (using semidefinite programming), the optimization might still be computationally demanding. Further, the algorithms has several parameters, whose influence remains unclear. This makes a judgement on the ease of use for practical problems difficult. Further, the theoretical analysis is very specific for this problem. As a general proof outline is somehow missing, it remains difficult to see how much of the analysis can transfer to other problems. In summary, the contributions are on a very specific topic. Minor Comments - line 103 thereore - therefore - line 383 problemmmm - Line 399: The forward reference to Appendix F is not nice. Can you make a preliminaries section for the Appendix? - Algorithm 1: What is ||K_*|| exactly? Is this an arbitrary constant or does the algorithm require to know the norm of the optimal controller correctly?

Reviewer 2



The authors propose a polynomial time algorithm for robust adaptive control of unknown linear systems (with quadratic objective functions). The algorithm provides a sublinear regret of O(T^{2/3}) with high probability. Moreover, a lower bound on the expected regret is provided by studying the relation between system parameter estimation and regret minimization problems. The paper is well written, the problem is relevant, and the authors succeed in effectively placing their work relative to the existing literature. I have to admit that I haven't been able to follow details of many of the technicals developments provided in the supplementary material. Furthermore, several issues should be addressed in the revised version. High-level comments: - It would be useful if the authors could please explain why regret provides a proper metric for quantifying performance of an algorithm with uncertain parameters. I recognize that several other references also use regret but I'm not sure why this is a proper metric for evaluating performance of controllers for uncertain systems. - The paper provides probabilistic guarantees. Since there are many sources of stochastic uncertainty, it would be useful if the authors could please clarify what feature of the problem leads to the results that hold with high probability. I would also like to understand how the choice of parameter \delta in the problem setup affects the algorithm. - The system-level synthesis appears to be a key technical enabler for the controls part of the paper. This approach leads to an infinite-dimensional synthesis problem and the authors go around this issue by solving a finite-dimensional approximation based on FIR. In contrast to standard LQR, the resulting controller appears to be dynamic and it is not clear how much conservatism FIR restriction introduces and how the FIR truncation length influences the performance. It would be useful if the authors could provide some additional comments about this. - Technical comments: - According to the Appendix G.3, the 8th line of the algorithm which is an infinite-dimensional problem can be brought into a finite-dimensional SDP using an approximation based on FIR. Since solving large SDPs is in general challenging, the computational complexity should be discussed. Please explain how such problems can be handled for large-scale systems. - While the OFU-based method can achieve an optimal regret in LQR problems, the authors state that there is no efficient algorithm for solving non-convex optimization problems in OFU. Along the same lines as in the previous comment, the authors should provide computational complexity analysis of their algorithm and compare it to OFU for a large-scale system. This is important mainly because of recent developments in solving the non-convex optimization subroutine encountered in OFU. On a related note, the paper would benefit from a larger numerical example in Section 4 to better compare the efficiency of various methods against one another. - Line 6 of the algorithm uses a least-square approach to find the estimates of the matrices A and B. However, this method does not necessarily provide a correct estimate of A (e.g., the estimate \hat{A} may turnout to be stable for an unstable system which may have a detrimental influence on the performance of the designed controller). Moreover, Theorem A.2 depends on the existence of (I + \hat{\Delta})^{-1}, where \hat{\Delta} is unknown. Since the authors utilize non-standard techniques based on recently introduced system-level synthesis, some additional discussion would be useful on how the tools from robust control theory lead to such an estimate. Please also comment on how this affects the practical aspect of the proposed approach and how one should determine the constants of Algorithm 1 (e.g. C_x, C_u, and \rho) when the dynamics are unknown. - Showing the performance of various algorithms against the number of iterations can be misleading, especially when the algorithms involve subroutines that require different computational complexity. In addition to Figure 1, please also provide performance curves as a function of computation time. - Please discuss sensitivity of the performance of the proposed algorithm to the choices of C_\eta, \gamma, and the FIR truncation length F. - It is stated that the outer minimization in Step 8 of the algorithm can be solved via bisection. For larger values of gamma, the inner minimization problem is relaxed meaning that the optimal objective of the inner minimization becomes smaller. However, the coefficient 1/(1-gamma) in the outer minimization increases. Hence, it is unclear why a bisection can find the optimal gamma. Minor issues: - How does the assumption of Gaussian noise influence the results (if at all)? - Line 226: please provide a reference for the heuristic projected gradient descent implementation of OFU. - Line 43: please define ’T’. - It would be useful to explain the choice of parameter \sigma_{\eta,i}^2 inside the algorithm as well. - Line 34: please explain what is meant by ‘PAC-learnable’. - Line 103: “thereore” should be “therefore”. - Line 181: "degredation" - Line 183: “,” should be “.”. - Line 383: "problemmmm"

Reviewer 3



The paper addresses adaptive control for the LQR problem. A polynomial time algorithm is proposed and shown to give sublinear (O(T^{2/3}) regret. This is claimed to be the first such result. A lower bound on the regret is also provided in terms of the system excitation, which is related to parameter estimation error. I view this paper as a solid theoretical paper, which provides novel results. The proposed algorithm appears to include novel features, and the results meaningful. (Unfortunately, I am not knowledgeable enough in the recent literature to say that with confidence.) The paper is generally well written. The interested reader who is not an expert will need to frequently consult the appendix, but this seems unavoidable. While the proposed algorithm and results are of theoretical significance, it may be pointed out that their practical importance is limited, as few systems are fully linear and time invariant.